# PRIORGRAD: IMPROVING CONDITIONAL DENOISING DIFFUSION MODELS WITH DATA-DEPENDENT ADAPTIVE PRIOR

**Sang-gil Lee**[1]* **Heeseung Kim**[1] **Chaehun Shin**[1] **Xu Tan**[2]† **Chang Liu**[2]

**Qi Meng**[2] **Tao Qin**[2] **Wei Chen**[2] **Sungroh Yoon**[1,3]† **Tie-Yan Liu**[2]

[1]Data Science & AI Lab., Seoul National University  [2]Microsoft Research Asia
[3] AIIS, ASRI, INMC, ISRC, NSI, and Interdisciplinary Program in Artificial Intelligence, Seoul National University
[1]{tkdrlf9202, gmltmd789, chaehuny, sryoon}@snu.ac.kr
[2]{xuta, changliu, meq, taoqin, wche, tyliu}@microsoft.com

## ABSTRACT

Denoising diffusion probabilistic models have been recently proposed to generate high-quality samples by estimating the gradient of the data density. The framework defines the prior noise as a standard Gaussian distribution, whereas the corresponding data distribution may be more complicated than the standard Gaussian distribution, which potentially introduces inefficiency in denoising the prior noise into the data sample because of the discrepancy between the data and the prior. In this paper, we propose PriorGrad to improve the efficiency of the conditional diffusion model for speech synthesis (for example, a vocoder using a mel-spectrogram as the condition) by applying an adaptive prior derived from the data statistics based on the conditional information. We formulate the training and sampling procedures of PriorGrad and demonstrate the advantages of an adaptive prior through a theoretical analysis. Focusing on the speech synthesis domain, we consider the recently proposed diffusion-based speech generative models based on both the spectral and time domains and show that PriorGrad achieves faster convergence and inference with superior performance, leading to an improved perceptual quality and robustness to a smaller network capacity, and thereby demonstrating the efficiency of a data-dependent adaptive prior.

## 1 INTRODUCTION

Deep generative models have been achieving rapid progress, by which deep neural networks approximate the data distribution and synthesize realistic samples from the model. There is a wide range of this type of approach, ranging from autoregressive models (Oord et al., 2016a;b), generative adversarial networks (Goodfellow et al., 2014; Brock et al., 2019), variational autoencoders (Kingma & Welling, 2013; Vahdat & Kautz, 2020), and normalizing flows (Rezende & Mohamed, 2015; Kingma & Dhariwal, 2018). Denoising diffusion probabilistic models (DDPMs) (Ho et al., 2020) and score matching (SM) (Song & Ermon, 2019) are recently proposed categories that can be used to synthesize high-fidelity samples with competitive or sometimes better quality than previous state-of-the-art approaches. Consequently, there have been a variety of applications based on DDPM or SM (Saharia et al., 2021; Kawar et al., 2021). Speech synthesis is one of the most successful applications, where the diffusion model can synthesize spectral or time-domain audio conditioned on text or spectral information, respectively, achieving a competitive quality but faster sampling (Chen et al., 2021; Kong et al., 2021; Jeong et al., 2021; Lee & Han, 2021) than autoregressive models (Oord et al., 2016b; Kalchbrenner et al., 2018).

---

*Work done during an internship at Microsoft Research Asia
†Corresponding Authors

However, although the diffusion-based speech synthesis models have achieved high-quality speech audio generation, they exhibit potential inefficiency, which may necessitate advanced strategies. For example, the model suffers from a significantly slow convergence during training, and a prohibitively large training computation time is required to learn the approximate reverse diffusion process. We investigate the diffuion-based models and observe the discrepancy between the real data distribution and the choice of the prior. Existing diffusion-based models define a standard Gaussian as the prior distribution and design a non-parametric diffusion process that procedurally destroys the signal into the prior noise. The deep neural network is trained to approximate the reverse diffusion process by estimating the gradient of the data density. Although applying the standard Gaussian as the prior is simple without any assumptions on the target data, it also introduces inefficiency. For example, in time-domain waveform data, the signal has extremely high variability between different segments such as voiced and unvoiced parts. Jointly modeling the voiced and unvoiced segments with the same standard Gaussian prior may be difficult for the model to cover all modes of the data, leading to training inefficiencies and potentially spurious diffusion trajectories.

Given the previous reasoning, we assessed the following question: *For a conditional diffusion-based model, can we formulate a more informative prior without incorporating additional computational or parameter complexity?* To investigate this, we propose a simple yet effective method, called PriorGrad, that uses adaptive noise by directly computing the mean and variance for the forward diffusion process prior, based on the conditional information. Specifically, using a conditional speech synthesis model, we propose structuring the prior distribution based on the conditional data, such as a mel-spectrogram for the vocoder (Chen et al., 2021; Kong et al., 2021) and a phoneme for the acoustic model (Jeong et al., 2021). By computing the statistics from the conditional data at the frame level (vocoder) or phoneme-level (acoustic model) granularity and mapping them as the mean and variance of the Gaussian prior, we can structure the noise that is similar to the target data distribution at an instance level, easing the burden of learning the reverse diffusion process.

We implemented PriorGrad based on the recently proposed diffusion-based speech generative models (Kong et al., 2021; Chen et al., 2021; Jeong et al., 2021), and conducted experiments on the LJSpeech (Ito & Johnson, 2017) dataset. The experimental results demonstrate the benefits of PriorGrad, such as a significantly faster model convergence during training, improved perceptual quality, and an improved tolerance to a reduction in network capacity. Our contributions are as follows:

- To the best of our knowledge, our study is one of the first to systematically investigate the effect of using a non-standard Gaussian distribution as the forward diffusion process prior to the conditional generative model.
- Compared to previous non-parametric forward diffusion without any assumption, we show that the model performance is significantly improved with faster convergence by leveraging the conditional information as the adaptive prior.
- We provide a comprehensive empirical study and analysis of the diffusion model behavior in speech generative models, in both the spectral and waveform domains, and demonstrate the effectiveness of the method, such as a significantly accelerated inference and improved quality.

## 2 BACKGROUND

In this section, we describe the basic formulation of the diffusion-based model and provide related studies, along with a description of our contribution with PriorGrad.

**Basic formulation** Denoising diffusion probabilistic models (DDPM) (Ho et al., 2020) are recently proposed deep generative models defined by two Markov chains: forward and reverse processes. The *forward process* procedurally destroys the data $\mathbf{x}_0$ into a standard Gaussian $\mathbf{x}_T$, as follows:

$$q(\mathbf{x}_{1:T}|\mathbf{x}_0) = \prod_{t=1}^{T} q(\mathbf{x}_t|\mathbf{x}_{t-1}), \quad q(\mathbf{x}_t|\mathbf{x}_{t-1}) := \mathcal{N}(\mathbf{x}_t; \sqrt{1-\beta_t}\mathbf{x}_{t-1}, \beta_t\mathbf{I}), \tag{1}$$

where $q(\mathbf{x}_t|\mathbf{x}_{t-1})$ represents the transition probability at the $t$-th step using a user-defined noise schedule $\beta_t \in \{\beta_1, ..., \beta_T\}$. Thus, the noisy distribution of $\mathbf{x}_t$ is the closed form of $q(\mathbf{x}_t|\mathbf{x}_0) =$

$\mathcal{N}(\mathbf{x}_t; \sqrt{\bar{\alpha}_t}\mathbf{x}_0, (1-\bar{\alpha}_t)\mathbf{I})$, where $\alpha_t := 1-\beta_t$, $\bar{\alpha}_t := \prod_{s=1}^{t}\alpha_s$. $q(\mathbf{x}_T|\mathbf{x}_0)$ converges in distribution to the standard Gaussian $\mathcal{N}(\mathbf{x}_T; \mathbf{0}, \mathbf{I})$ if $\bar{\alpha}_T$ is small enough based on a carefully designed noise schedule. The *reverse process* that procedurally transforms the prior noise into data is defined as follows:

$$p_\theta(\mathbf{x}_{0:T}) = p(\mathbf{x}_T)\prod_{t=1}^{T} p_\theta(\mathbf{x}_{t-1}|\mathbf{x}_t), \quad p_\theta(\mathbf{x}_{t-1}|\mathbf{x}_t) = \mathcal{N}\left(\mathbf{x}_{t-1}; \boldsymbol{\mu}_\theta(\mathbf{x}_t, t), \boldsymbol{\Sigma}_\theta(\mathbf{x}_t, t)\right), \quad (2)$$

where $p(\mathbf{x}_T) = \mathcal{N}(\mathbf{x}_T; \mathbf{0}, \mathbf{I})$ and $p_\theta(\mathbf{x}_{t-1}|\mathbf{x}_t)$ corresponds to the reverse of the forward transition probability, parameterized using a deep neural network. We can define the evidence lower bound (ELBO) loss as the training objective of the reverse process:

$$L(\theta) = \mathbb{E}_q\left[\mathrm{KL}\left(q(\mathbf{x}_T|\mathbf{x}_0)||p(\mathbf{x}_T)\right) + \sum_{t=2}^{T}\mathrm{KL}(q(\mathbf{x}_{t-1}|\mathbf{x}_t, \mathbf{x}_0)||p_\theta(\mathbf{x}_{t-1}|\mathbf{x}_t)) - \log p_\theta(\mathbf{x}_0|\mathbf{x}_1)\right].$$
$$(3)$$

As shown in Ho et al. (2020), $q(\mathbf{x}_{t-1}|\mathbf{x}_t, \mathbf{x}_0)$ can be represented by Bayes rule as follows:

$$q(\mathbf{x}_{t-1}|\mathbf{x}_t, \mathbf{x}_0) = N(\mathbf{x}_{t-1}; \tilde{\mu}(\mathbf{x}_t, \mathbf{x}_0), \tilde{\beta}_t\mathbf{I}), \quad (4)$$

$$\tilde{\boldsymbol{\mu}}_t(\mathbf{x}_t, \mathbf{x}_0) := \frac{\sqrt{\bar{\alpha}_{t-1}}\beta_t}{1-\bar{\alpha}_t}\mathbf{x}_0 + \frac{\sqrt{\alpha_t}(1-\bar{\alpha}_{t-1})}{1-\bar{\alpha}_t}\mathbf{x}_t, \quad \tilde{\beta}_t := \frac{1-\bar{\alpha}_{t-1}}{1-\bar{\alpha}_t}\beta_t. \quad (5)$$

By fixing $p(\mathbf{x}_T)$ as a standard Gaussian, $\mathrm{KL}(q(\mathbf{x}_T|\mathbf{x}_0)||p(\mathbf{x}_T))$ becomes constant and is not parameterized. The original framework in Ho et al. (2020) fixed $\boldsymbol{\Sigma}_\theta(\mathbf{x}_t, t)$ as a constant $\tilde{\beta}_t\mathbf{I}$ and set the standard Gaussian noise $\epsilon$ as the optimization target instead of $\tilde{\mu}_t$ by reparameterizing $\mathbf{x}_0 = \frac{1}{\sqrt{\bar{\alpha}_t}}(\mathbf{x}_t - \sqrt{1-\bar{\alpha}_t}\epsilon)$ from $q(\mathbf{x}_t|\mathbf{x}_0)$ to minimize the second and third terms in equation 3. Based on this setup, in Ho et al. (2020), the authors further demonstrated that we can drop the weighting factor of each term and use a simplified training objective that provides a higher sample quality:

$$-\mathrm{ELBO} = C + \sum_{t=1}^{T}\mathbb{E}_{\mathbf{x}_0, \epsilon}\left[\frac{\beta_t^2}{2\sigma_t^2\alpha_t(1-\bar{\alpha}_t)}\|\epsilon - \epsilon_\theta(\sqrt{\bar{\alpha}_t}\mathbf{x}_0 + \sqrt{1-\bar{\alpha}_t}\epsilon, t)\|^2\right], \quad (6)$$

$$L_{\mathrm{simple}}(\theta) := \mathbb{E}_{t, \mathbf{x}_0, \epsilon}\left[\|\epsilon - \epsilon_\theta(\mathbf{x}_t, t)\|^2\right]. \quad (7)$$

**Related work** Since the introduction of the DDPM, there have been a variety of further studies (Nichol & Dhariwal, 2021; Song et al., 2020), applications (Chen et al., 2021; Jeong et al., 2021; Kong et al., 2021; Saharia et al., 2021), and a symbiosis of diffusion (Ho et al., 2020; Sohl-Dickstein et al., 2015) and score-based models (Song & Ermon, 2019; 2020) as a unified view with stochastic differential equations (SDEs) (Song et al., 2021). From an application perspective, several conditional generative models have been proposed. Waveform synthesis models (Chen et al., 2021; Kong et al., 2021) are one of the major applications in which the diffusion model is trained to generate time-domain speech audio from the prior noise, conditioned on a mel-spectrogram. A diffusion-based decoder has also been applied to text-to-spectrogram generation models (Jeong et al., 2021; Popov et al., 2021). PriorGrad focuses on improving the efficiency of training such methods from the perspective of a conditional generative model. We investigate the potential inefficiency of the current methods which require unfeasibly large computing resources to train and generate high-quality samples.

Studies on formulating an informative prior distribution for deep generative model are not new, and there has been a variety of studies investigating a better prior, ranging from hand-crafted (Nalisnick & Smyth, 2017; Tomczak & Welling, 2018), autoregressive (Chen et al., 2017), vector quantization (Razavi et al., 2019), prior encoder (Rezende & Mohamed, 2015), and data-dependent approaches similar to ours (Li et al., 2019). We tackle the problem of training inefficiency of diffusion-based models by crafting better priors in a data-dependent manner, where our method can provide a better trajectory and can reduce spurious modes, enabling more efficient training. Nachmani et al. (2021)

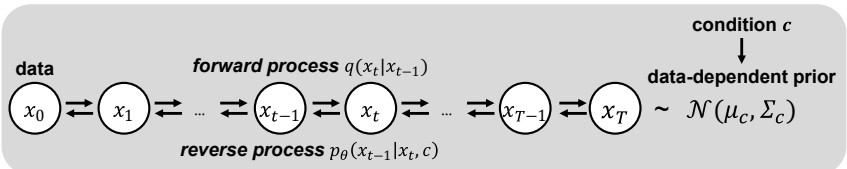

Figure 1: High-level overview of the proposed method with the directed graphical model.

used Gamma distribution as the diffusion prior. Note that there has also been a concurrent study conducted on leveraging the prior distribution on the acoustic model, Grad-TTS (Popov et al., 2021), in which the effectiveness of using the mean-shifted Gaussian as a prior with the identity variance was investigated. Unlike the method in Popov et al. (2021), which enforces the encoder output to match the target mel-spectrogram by using an additional encoder loss, our approach augments the forward diffusion prior directly through data and the encoder has no restriction on latent feature representations. In Popov et al. (2021), the forward diffusion prior is jointly trained and may induce additional overhead on convergence as the prior changes throughout the training, whereas our method provides guaranteed convergence through the fixed informative prior.

## 3 METHOD

We investigate the following intuitive argument: *When we structure the informative prior noise closer to the data distribution, can we improve the efficiency of the diffusion model?* In this section, we present a general formulation of PriorGrad, describe training and sampling algorithms, and provide benefits of PriorGrad through a theoretical analysis. PriorGrad offers a generalized approach for the diffusion-based model with the non-standard Gaussian as the prior and can be applied to a variety of applications.

### 3.1 GENERAL FORMULATION

In this section, we provide a general formulation of the method regarding using the non-standard Gaussian $\mathcal{N}(\boldsymbol{\mu}, \boldsymbol{\Sigma})$ as the forward diffusion prior. PriorGrad leverages the conditional data to directly compute instance-level approximate priors in an adaptive manner, and provides the approximate prior as the forward diffusion target for both training and inference. Figure 1 and 2 presents a visual high-level overview. We take the same parameterization of $\boldsymbol{\mu}_\theta$ and $\boldsymbol{\sigma}_\theta$ as in the original DDPM (Ho et al., 2020) from Section 2, as follows:

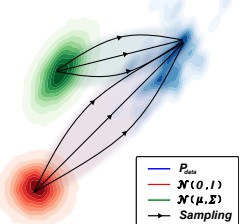

$$\boldsymbol{\mu}_\theta(\mathbf{x}_t, t) = \frac{1}{\sqrt{\alpha_t}}\left(\mathbf{x}_t - \frac{\beta_t}{\sqrt{1-\bar{\alpha}_t}}\boldsymbol{\epsilon}_\theta(\mathbf{x}_t, t)\right), \quad \sigma_\theta(\mathbf{x}_t, t) = \tilde{\beta}_t^{\frac{1}{2}} \quad (8)$$

Figure 2: Illustrative description of the diffusion trajectory with the non-standard Gaussian.

Assuming that we have access to an optimal Gaussian $\mathcal{N}(\boldsymbol{\mu}, \boldsymbol{\Sigma})$ as the forward diffusion prior distribution, we have the following modified ELBO:

**Proposition 1** *Let $\boldsymbol{\epsilon} \sim \mathcal{N}(\mathbf{0}, \boldsymbol{\Sigma})$ and $\mathbf{x}_0 \sim q_{data}$. Then, under the parameterization in Eq. equation 8, the ELBO loss will be*

$$-\text{ELBO} = C(\boldsymbol{\Sigma}) + \sum_{t=1}^{T} \gamma_t \mathbb{E}_{\mathbf{x}_0, \boldsymbol{\epsilon}} \|\boldsymbol{\epsilon} - \boldsymbol{\epsilon}_\theta(\sqrt{\bar{\alpha}_t}(\mathbf{x}_0 - \mu) + \sqrt{1-\bar{\alpha}_t}\boldsymbol{\epsilon}, t)\|^2_{\boldsymbol{\Sigma}^{-1}},$$

*for some constant C, where $\|\mathbf{x}\|^2_{\boldsymbol{\Sigma}^{-1}} = \mathbf{x}^T \boldsymbol{\Sigma}^{-1} \mathbf{x}$, $\gamma_t = \frac{\beta_t^2}{2\sigma_t^2 \alpha_t (1-\bar{\alpha}_t)}$ for $t > 1$, and $\gamma_1 = \frac{1}{2\alpha_1}$.*

The Proposition 1 is an extension of the ELBO in Equation 6 with the non-standard Gaussian distribution. Contrary to original DDPM, which used $\mathcal{N}(\mathbf{0}, \mathbf{I})$ as the prior without any assumption on data, through Proposition 1, we train the model with $\mathcal{N}(\boldsymbol{\mu}, \boldsymbol{\Sigma})$, whose mean and variance are extracted from the data, as the prior for the forward process. See appendix A.1 for full derivation. We

also drop $\gamma_t$ as the simplified loss $\mathcal{L} = \|\boldsymbol{\epsilon} - \boldsymbol{\epsilon}_\theta(\mathbf{x}_t, c, t)\|_{\boldsymbol{\Sigma}^{-1}}^2$ for training, following the previous work. Algorithms 1 and 2 describe the training and sampling procedures augmented by the data-dependent prior $(\boldsymbol{\mu}, \boldsymbol{\Sigma})$. Because computing the data-dependent prior is application-dependent, in Section 4 and 5, we describe how to compute such prior based on the conditional data on the given task.

---

**Algorithm 1** Training of PriorGrad

> **repeat**
>   $(\boldsymbol{\mu}, \boldsymbol{\Sigma})$ = data-dependent prior
>   Sample $\mathbf{x}_0 \sim q_{data}, \boldsymbol{\epsilon} \sim \mathcal{N}(\mathbf{0}, \boldsymbol{\Sigma})$
>   Sample $t \sim \mathcal{U}(\{1, \cdots, T\})$
>   $\mathbf{x}_t = \sqrt{\bar{\alpha}_t}(\mathbf{x}_0 - \boldsymbol{\mu}) + \sqrt{1 - \bar{\alpha}_t}\boldsymbol{\epsilon}$
>   $\mathcal{L} = \|\boldsymbol{\epsilon} - \boldsymbol{\epsilon}_\theta(\mathbf{x}_t, c, t)\|_{\boldsymbol{\Sigma}^{-1}}^2$
>   Update the model parameter $\theta$ with $\nabla_\theta \mathcal{L}$
> **until** converged

**Algorithm 2** Sampling of PriorGrad

> $(\boldsymbol{\mu}, \boldsymbol{\Sigma})$ = data-dependent prior
> Sample $\mathbf{x}_T \sim \mathcal{N}(\mathbf{0}, \boldsymbol{\Sigma})$
> **for** $t = T, T-1, \cdots, 1$ **do**
>   Sample $\mathbf{z} \sim \mathcal{N}(\mathbf{0}, \boldsymbol{\Sigma})$ if $t > 1$; else $\mathbf{z} = \mathbf{0}$
>   $\mathbf{x}_{t-1} = \frac{1}{\sqrt{\alpha_t}}(\mathbf{x}_t - \frac{1-\alpha_t}{\sqrt{1-\bar{\alpha}_t}}\boldsymbol{\epsilon}_\theta(\mathbf{x}_t, c, t)) + \sigma_t \mathbf{z}$
> **end for**
> **return** $\mathbf{x}_0 + \boldsymbol{\mu}$

---

### 3.2 THEORETICAL ANALYSIS

In this section, we describe the theoretical benefits of PriorGrad. First, we discuss about the simplified modeling with the following proposition:

**Proposition 2** *Let $L(\boldsymbol{\mu}, \boldsymbol{\Sigma}, \mathbf{x}_0; \theta)$ denote the $-$ELBO loss in Proposition 1. Suppose that $\boldsymbol{\epsilon}_\theta$ is a linear function. Under the constraint that $\det(\boldsymbol{\Sigma}) = \det(\mathbf{I})$, we have $\min_\theta L(\boldsymbol{\mu}, \boldsymbol{\Sigma}, \mathbf{x}_0; \theta) \leq \min_\theta L(\mathbf{0}, \mathbf{I}, \mathbf{x}_0; \theta)$.*

The proposition shows that setting the prior whose covariance $\boldsymbol{\Sigma}$ aligns with the covariance of data $\mathbf{x}_0$ leads to a smaller loss if we use a linear function approximation for $\boldsymbol{\epsilon}_\theta$. This indicates that we can use a simple model to represent the mean of $q(\mathbf{x}_{t-1}|\mathbf{x}_t)$ under the data-dependent prior, whereas we need to use a complex model to achieve the same precision under a prior with isotropic covariance. The condition $\det(\boldsymbol{\Sigma}) = \det(\mathbf{I})$ means that the two Gaussian priors have equal entropy, which is a condition for a fair comparison.

Second, we discuss the convergence rate. The convergence rate of optimization depends on the condition number of the Hessian matrix of the loss function (denoted as $\mathbf{H}$) (Nesterov, 2003), that is, $\frac{\lambda_{\max}(\mathbf{H})}{\lambda_{\min}(\mathbf{H})}$, where $\lambda_{\max}$ and $\lambda_{\min}$ are the maximal and minimal eigenvalues of $\mathbf{H}$, respectively. A smaller condition number leads to a faster convergence rate. For $L(\boldsymbol{\mu}, \boldsymbol{\Sigma}, \mathbf{x}_0; \theta)$, the Hessian is calculated as

$$\mathbf{H} = \frac{\partial^2 L}{\partial \boldsymbol{\epsilon}_\theta^2} \cdot \frac{\partial \boldsymbol{\epsilon}_\theta}{\partial \theta} \cdot \left(\frac{\partial \boldsymbol{\epsilon}_\theta}{\partial \theta}\right)^T + \frac{\partial L}{\partial \boldsymbol{\epsilon}_\theta} \cdot \frac{\partial^2 \boldsymbol{\epsilon}_\theta}{\partial \theta^2} \tag{9}$$

Again, if we assume $\boldsymbol{\epsilon}_\theta$ is a linear function, we have $\mathbf{H} \propto \mathbf{I}$ for $L(\boldsymbol{\mu}, \boldsymbol{\Sigma}, \mathbf{x}_0; \theta)$ and $\mathbf{H} \propto \boldsymbol{\Sigma} + \mathbf{I}$ for $L(\mathbf{0}, \mathbf{I}, \mathbf{x}_0; \theta)$. It is clear that if we set the prior to be $\mathcal{N}(\boldsymbol{\mu}, \boldsymbol{\Sigma})$, the condition number of $\mathbf{H}$ equals 1, achieving the smallest value of the condition number. Therefore, it can accelerate the convergence. Readers may refer to the appendix A.2 for more details.

## 4 APPLICATION TO VOCODER

In this section, we apply PriorGrad to a vocoder model, as visually described in Figure 3.

### 4.1 PRIORGRAD FOR VOCODER

Our formulation of PriorGrad applied to a vocoder is based on DiffWave (Kong et al., 2021), where the model synthesizes time-domain waveform conditioned on a mel-spectrogram that contains a compact frequency feature representation of the data. Unlike the previous method that used $\boldsymbol{\epsilon} \sim \mathcal{N}(\mathbf{0}, \mathbf{I})$, PriorGrad network is trained to estimate the noise $\boldsymbol{\epsilon} \sim \mathcal{N}(\mathbf{0}, \boldsymbol{\Sigma})$ given the destroyed signal $\sqrt{\bar{\alpha}_t}\mathbf{x}_0 + \sqrt{1 - \bar{\alpha}_t}\boldsymbol{\epsilon}$. Same as Kong et al. (2021), the network is also conditioned on the discretized

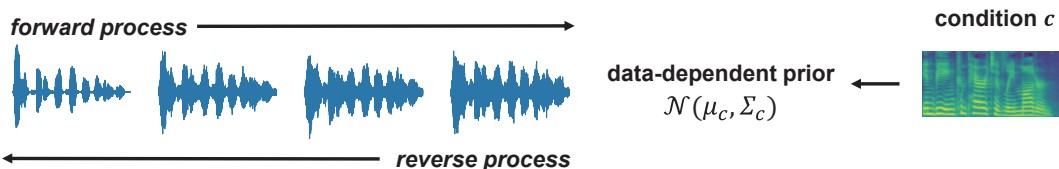

Figure 3: Visual description of PriorGrad for vocoder.

index of the noise level $\sqrt{\bar{\alpha}_t}$ with the diffusion-step embedding layers, and the mel-spectrogram $c$ with the conditional projection layers.

Based on the mel-spectrogram condition, we propose leveraging a normalized frame-level energy of the mel-spectrogram for acquiring data-dependent prior, exploiting the fact that the spectral energy contains an exact correlation to the waveform variance[1]. First, we compute the frame-level energy by applying roots of sum of exponential to $\mathbf{c}$, where $\mathbf{c}$ is the mel-spectrogram from the training dataset. We then normalize the frame-level energy to a range of $(0, 1]$ to acquire the data-dependent diagonal variance $\Sigma_\mathbf{c}$. In this way, we can use the frame-level energy as a proxy of the standard deviation for the waveform data we want to model. This can be considered as a non-linear mapping between the mel-scale spectral energy and the standard deviation of the diagonal Gaussian.

We set $\mathcal{N}(\mathbf{0}, \Sigma)$ as the forward diffusion prior for each training step by upsampling $\Sigma_\mathbf{c}$ in the frame-level to $\Sigma$ in the waveform-level using a hop length for the given vocoder. We chose a zero-mean prior in this setup reflecting the fact that the waveform distribution is zero-mean. In practice, we imposed the minimum standard deviation of the prior to $0.1$ through clipping to ensure numerical stability during training[2]. We have tried several alternative sources of conditional information to compute the prior, such as voiced/unvoiced labels and phoneme-level statistics, but they resulted in a worse performance. We justify our choice of using frame-level energy in the appendix A.3.

## 4.2 EXPERIMENTAL SETUP

We used LJSpeech (Ito & Johnson, 2017) dataset for all experiments, which is a commonly used open-source 24h speech dataset with 13,100 audio clips from a single female speaker. We used an 80-band mel-spectrogram feature at the log scale from the 22,050Hz volume-normalized speech, with the 1024-point FFT, 80Hz and 7,600Hz low- and high-frequency cutoff, and a hop length of 256. We used 13,000 clips as the training set, 5 clips as the validation set, and the remaining 95 clips as the test set used for an objective and subjective audio quality evaluation. We followed the publicly available implementation[3], where it uses a 2.62M parameter model with an Adam optimizer (Kingma & Ba, 2014) and a learning rate of $2 \times 10^{-4}$ for a total of 1M iterations. Training for 1M iterations took approximately 7 days with a single NVIDIA A40 GPU. We used the default diffusion steps with $T = 50$ and the linear beta schedule ranging from $1 \times 10^{-4}$ to $5 \times 10^{-2}$ for training and inference, which is the default setting of DiffWave$_{BASE}$ model used in Kong et al. (2021). We also used the fast $T_{infer} = 6$ inference noise schedule from DiffWave$_{BASE}$ without modification.

## 4.3 EXPERIMENTAL RESULTS

We conducted experiments to verify whether our method can learn the reverse diffusion process faster, by comparing to the baseline DiffWave model with $\epsilon \sim \mathcal{N}(\mathbf{0}, \mathbf{I})$. First, we present the model convergence result by using a spectral domain loss on the test set to show the fast training of PriorGrad. Second, we provide both objective and subjective audio quality results, where PriorGrad offered the improved quality. Finally, we measure a tolerance to the reduction of the network capacity, where PriorGrad exhibits an improved parameter efficiency. We leave the description of the objective metrics we collected to the appendix A.4.

---

[1]This property about the spectral density is dictated by Parseval's theorem (Stoica et al., 2005).

[2]In our preliminary study, we applied grid search over the minimum standard deviation from 0.1 to 0.0001, and it showed negligible difference as long as the minimum is clipped to the reasonably low value. Not clipping it entirely can result in numerical instability.

[3]https://github.com/lmnt-com/diffwave

Table 1: 5-scale subjective mean opinion score (MOS) results of PriorGrad vocoder with 95% confidence intervals.

Figure 4: Model convergence result of vocoder models measured by log-mel spectrogram mean absolute error (LS-MAE).

| Method | $T_{infer}$ | Training Steps | |
| --- | --- | --- | --- |
| | | 500K | 1M |
| GT | - | $4.42 \pm 0.07$ | |
| DiffWave | 6 | $3.98 \pm 0.08$ | $4.01 \pm 0.08$ |
| | 50 | $4.12 \pm 0.08$ | $4.12 \pm 0.08$ |
| PriorGrad | 6 | $4.02 \pm 0.08$ | $4.14 \pm 0.08$ |
| | 50 | $4.21 \pm 0.08$ | $\mathbf{4.25 \pm 0.08}$ |

Table 2: MOS results of PriorGrad vocoder under reduced model capacity, evaluated at 1M training step.

| Method | Parameters | |
| --- | --- | --- |
| | Base (2.62M) | Small (1.23M) |
| GT | $4.38 \pm 0.08$ | |
| DiffWave | $4.06 \pm 0.08$ | $3.90 \pm 0.09$ |
| PriorGrad | $\mathbf{4.12 \pm 0.08}$ | $\mathbf{4.02 \pm 0.08}$ |

Table 3: Objective metric results of PriorGrad vocoder at 1M training steps. Lower is better.

| Method | LS-MAE ($\downarrow$) | MR-STFT ($\downarrow$) | MCD ($\downarrow$) | $F_0$ RMSE ($\downarrow$) | $S(x_T, x_0)$ ($\downarrow$) | $S(\tilde{x}_0, x_0)$ ($\downarrow$) |
| --- | --- | --- | --- | --- | --- | --- |
| DiffWave | 0.5264 | 1.0920 | 9.7822 | 16.4035 | 72698.62 | 1650.22 |
| PriorGrad | **0.5048** | **0.9976** | **9.2820** | **15.5542** | **42236.93** | **1608.89** |

**Model convergence** We used a widely adopted spectral metric with log-mel spectrogram mean absolute error (LS-MAE) as the proxy of the convergence for the waveform synthesis model. We can see from Figure 4 that PriorGrad exhibited a significantly faster spectral convergence compared to the baseline. In the auditory test, we observed that PriorGrad readily removed the background white noise early in training, whereas the baseline needed to learn the entire reverse diffusion process starting from $\epsilon \sim \mathcal{N}(\mathbf{0}, \mathbf{I})$ which contains little information.

Table 1 shows the 5-scale subjective mean opinion score (MOS) test of PriorGrad from Amazon Mechanical Turk. We observed that PriorGrad outperformed the baseline DiffWave model with 2 times fewer training iterations. PriorGrad also outperformed the baseline on objective speech and distance metrics as shown in Table 3, such as Multi-resolution STFT error (MR-STFT) (Yamamoto et al., 2020), Mel cepstral distortion (MCD) (Kubichek, 1993) and $F_0$ root mean square error ($F_0$ RMSE) and a debiased Sinkhorn divergence (Feydy et al., 2019) between the prior and the ground-truth $S(x_T, x_0)$, or the generated samples and the ground-truth $S(\tilde{x}_0, x_0)$, consistent with the subjective quality results. Refer to A.7 for comparison to other state-of-the-art approaches with varying number of inference steps.

**Parameter efficiency** We measured a tolerance to a reduced diffusion network capacity by setting the width of the dilated convolutional layers by half, leading to a smaller model with 1.23M parameters. We trained the small DiffWave and PriorGrad with the same training configurations for 1M steps. With $T_{infer} = 50$, Table 2 confirmed that the performance degradation of PriorGrad is reduced compared to the baseline DiffWave model. The small PriorGrad was able to maintain the quality close to the larger baseline model, suggesting that having access to the informative prior distribution can improve parameter efficiency of the diffusion-based generative model.

## 5 APPLICATION TO ACOUSTIC MODEL

In this section, we present the application of PriorGrad to the acoustic models, as visually described in Figure 5.

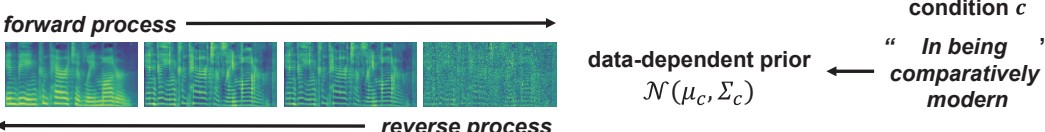

Figure 5: Visual description of PriorGrad for acoustic model.

## 5.1 PRIORGRAD FOR ACOUSTIC MODEL

The acoustic models generate a mel-spectrogram given a sequence of phonemes with the encoder-decoder architecture. In other words, the acoustic model is analogous to the text-conditional image generation task (Reed et al., 2016; Ramesh et al., 2021), where the target image is 2D mel-spectrogram. We implement the PriorGrad acoustic model by using the approach in Ren et al. (2020) as a feed-forward phoneme encoder, and using a diffusion-based decoder with dilated convolutional layers based on Kong et al. (2021). Note that a similar decoder architecture was used in Jeong et al. (2021).

To build the adaptive prior, we compute the phoneme-level statistics of the 80-band mel-spectrogram frames by aggregating the frames that correspond to the same phoneme from the training data, where phoneme-to-frame alignment is provided by the Montreal forced alignment (MFA) toolkit (McAuliffe et al., 2017). Specifically, for each phoneme, we acquire the 80-dimensional diagonal mean and variance from the aggregation of all occurrences in the training dataset. Then, we construct the dictionary of $\mathcal{N}(\boldsymbol{\mu}, \boldsymbol{\Sigma})$ per phoneme. To use these statistics for the forward diffusion prior, we need to upsample the phoneme-level prior sequence to the frame-level with the matching duration. This can be done by jointly upsampling this phoneme-level prior as same as the phoneme encoder output based on the duration predictor (Ren et al., 2019) module.

Following Algorithm 1, we train the diffusion decoder network with the mean-shifted noisy mel-spectrogram $\mathbf{x}_t = \sqrt{\bar{\alpha}_t}(\mathbf{x}_0 - \boldsymbol{\mu}) + \sqrt{1 - \bar{\alpha}_t}\boldsymbol{\epsilon}$ as the input. The network estimates the injected noise $\boldsymbol{\epsilon} \sim \mathcal{N}(\mathbf{0}, \boldsymbol{\Sigma})$ as the target. The network is additionally conditioned on the aligned phoneme encoder output. The encoder output is added as a bias term of each dilated convolution layers of the gated residual block with the layer-wise $1 \times 1$ convolution.

## 5.2 EXPERIMENTAL SETUP

For the acoustic model experiments, we implemented the Transformer phoneme encoder architecture identical to FastSpeech 2 (Ren et al., 2020) and the convolutional diffusion decoder architecture based on Jeong et al. (2021). We adopted the open-source implementation of the DiffWave architecture (Kong et al., 2021) with 12 convolutional layers, same as Jeong et al. (2021). We used the beta schedule with $T = 400$ for training and used the fast reverse sampling schedule with $T_{infer} = 6$ with a grid search method (Chen et al., 2021). We followed the same training and inference protocols of Ren et al. (2020) and used a pre-trained Parallel WaveGAN (PWG) (Yamamoto et al., 2020) vocoder for our controlled experiments. For state-of-the-art comparison, we used HiFi-GAN (Kong et al., 2020) vocoder to match the previous work. We conducted a comparative study of PriorGrad acoustic model with a different diffusion decoder network capacity, i.e., a small model with 3.5M parameters (128 residual channels), and a large model with 10M parameters (256 residual channels). Training for 300K iterations took approximately 2 days on a single NVIDIA P100 GPU. We leave the additional details in the appendix A.5.

## 5.3 EXPERIMENTAL RESULTS

**Model convergence** We can see from Table 4 that applying $\mathcal{N}(\boldsymbol{\mu}, \boldsymbol{\Sigma})$ with PriorGrad significantly accelerated the training convergence and exhibited higher-quality speech for different decoder capacities and training iterations. A high-capacity (10M) PriorGrad was able to generate high-quality mel-spectrogram with as few as 60K training iteration, whereas the baseline model was not able to match the performance even after 300K iterations. This is because the training iterations were insufficient for the baseline to learn the entire diffusion trajectory with the standard Gaussian as the prior, leading to lower performance. The small (3.5M) baseline model exhibited an improved

Table 4: MOS results of PriorGrad acoustic model with 95% confidence interval. We used a pre-trained Parallel WaveGAN (Yamamoto et al., 2020) for the vocoder.

| Method | Parameters (Decoder) | Training Steps | |
|---|---|---|---|
| | | 60K | 300K |
| GT | - | $4.41 \pm 0.08$ | |
| GT (Vocoder) | - | $4.22 \pm 0.08$ | |
| Baseline | 10M | $3.84 \pm 0.10$ | $3.91 \pm 0.09$ |
| PriorGrad | 10M | $4.04 \pm 0.07$ | $\mathbf{4.09 \pm 0.08}$ |
| Baseline | 3.5M | $3.87 \pm 0.09$ | $4.00 \pm 0.08$ |
| PriorGrad | 3.5M | $3.96 \pm 0.07$ | $\mathbf{4.08 \pm 0.07}$ |

quality compared to the large baseline for the same training iterations, which further suggests that the convergence of the large baseline model is slow. By contrast, by easing the burden of learning the diffusion process by having access to the phoneme-level informative forward prior, PriorGrad achieved high-quality samples significantly earlier in the training. Refer to Appendix A.6 for additional experimental results including an alternative model with the jointly trainable diffusion prior estimation, and Appendix A.7 for an expanded comparison to previous work, where PriorGrad achieves a new state-of-the-art acoustic model.

**Parameter efficiency** Similar to the vocoder experiment, we assessed whether we can retain the high performance with a reduced diffusion decoder capacity from 10M to 3.5M parameters. The small PriorGrad also consistently outperformed the baseline model and achieved an almost identical perceptual quality to the high-capacity PriorGrad. The small PriorGrad outperformed both the small and large baseline models, which confirms that PriorGrad offers an improved parameter efficiency and tolerance to the reduced network capacity. This suggests that PriorGrad provides a way to build practical and efficient diffusion-based generative models. This holds a special importance for speech synthesis models, where the efficiency plays a key role in model deployment in realistic environments.

## 6 DISCUSSION AND CONCLUSION

In this study, we investigated the potential inefficiency of recently proposed diffusion-based model as a conditional generative model within the speech synthesis domain. Our method, PriorGrad, directly leverages the rich conditional information and provides an instance-level non-standard adaptive Gaussian as the prior of the forward diffusion process. Through extensive experiments with the recently proposed diffusion-based speech generative models, we showed that PriorGrad achieves a faster model convergence, better denoising of the white background noise, an improved perceptual quality, and parameter efficiency. This enables the diffusion-based generative models to be significantly more practical, and real-world applications favor lightweight and efficient models for deployments.

Whereas our focus is on improving the diffusion-based speech generative models, PriorGrad has a potential to expand the application beyond the speech synthesis domain. In the image domain, for example, PriorGrad can potentially be applied to image super-resolution tasks (Saharia et al., 2021) by using the patch-wise mean and variance of the low-resolution image to dynamically control the forward/reverse process. The method could potentially be used for depth map conditional image synthesis (Esser et al., 2020), where the depth map can be mapped as the prior variance using the fact that the monocular depth corresponds to the camera focus and visual fidelity between the foreground and background. We leave these potential avenue of PriorGrad for future work.

PriorGrad also has limitations that require further investigation. Although PriorGrad offers a variety of practical benefits as presented, it may also require a well-thought out task-specific design to compute the data-dependent statistics (or its proxy), which may be unsuitable depending on the granularity of the conditional information. Building an advanced approach to enable a more generalized realization of PriorGrad will be an interesting area of future research.

ACKNOWLEDGEMENT

This work was supported by the BK21 FOUR program of the Education and Research Program for Future ICT Pioneers, Seoul National University in 2021, Institute of Information & communications Technology Planning & Evaluation (IITP) grant funded by the Korea government(MSIT) [NO.2021-0-01343, Artificial Intelligence Graduate School Program (Seoul National University)], and the MSIT (Ministry of Science, ICT), Korea, under the High-Potential Individuals Global Training Program (2020-0-01649) supervised by the IITP (Institute for Information & Communications Technology Planning & Evaluation), and Microsoft Research Asia.

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

# A APPENDIX

## A.1 DERIVATION OF THE ELBO LOSS

**Proposition 1** *Let* $\epsilon \sim \mathcal{N}(\mathbf{0}, \mathbf{\Sigma})$ *and* $\mathbf{x}_0 \sim q_{data}$*. Then, under the parameterization in Eq. equation 8, the ELBO loss will be*

$$- \mathrm{ELBO} = C(\mathbf{\Sigma}) + \sum_{t=1}^{T} \gamma_t \mathbb{E}_{\mathbf{x}_0, \epsilon} \| \epsilon - \epsilon_\theta(\sqrt{\bar{\alpha}_t}(\mathbf{x}_0 - \mu) + \sqrt{1 - \bar{\alpha}_t}\epsilon, t) \|_{\mathbf{\Sigma}^{-1}}^2,$$

*for some constant* $C$*, where* $\|\mathbf{x}\|_{\mathbf{\Sigma}^{-1}}^2 = \mathbf{x}^T \mathbf{\Sigma}^{-1} \mathbf{x}$*,* $\gamma_t = \frac{\beta_t^2}{2\sigma_t^2 \alpha_t (1 - \bar{\alpha}_t)}$ *for* $t > 1$*, and* $\gamma_1 = \frac{1}{2\alpha_1}$*.*

*Proof:*

According to Algorithm 1, the input $\mathbf{x}_0$ is normalized by subtracting its mean $\mu$. In the following, we denote $\tilde{\mathbf{x}}_0 = \mathbf{x}_0 - \mu$ and $\mathbf{x}_t = \sqrt{\bar{\alpha}_t}\tilde{\mathbf{x}}_0 + \sqrt{1 - \bar{\alpha}_t}\epsilon$.

According to Equation (3), the ELBO loss is

$$\mathrm{ELBO} = -\mathbb{E}_q \left( \mathrm{KL}(q(\mathbf{x}_T|\tilde{\mathbf{x}}_0)\|p(\mathbf{x}_T)) + \sum_{t=2}^{T} \mathrm{KL}(q(\mathbf{x}_{t-1}|\mathbf{x}_t, \tilde{\mathbf{x}}_0)\|p_\theta(\mathbf{x}_{t-1}|\mathbf{x}_t)) - \log p_\theta(\tilde{\mathbf{x}}_0|\mathbf{x}_1) \right).$$

Let $\epsilon_i$'s $\overset{i.i.d.}{\sim} \mathcal{N}(\mathbf{0}, \mathbf{\Sigma})$. Similarly to the Equation (1), we have $q(\mathbf{x}_t|\tilde{\mathbf{x}}_0) = \mathcal{N}(\mathbf{x}_t; \sqrt{\bar{\alpha}_t}\tilde{\mathbf{x}}_0, (1 - \bar{\alpha}_t)\mathbf{\Sigma})$.

By Bayes rule and Markov chain property,

$$q(\mathbf{x}_{t-1}|\mathbf{x}_t, \tilde{\mathbf{x}}_0) = \frac{q(\mathbf{x}_t|\mathbf{x}_{t-1})q(\mathbf{x}_{t-1}|\tilde{\mathbf{x}}_0)}{q(\mathbf{x}_t|\tilde{\mathbf{x}}_0)}$$

$$= \frac{\mathcal{N}(\mathbf{x}_t; \sqrt{\alpha_t}\mathbf{x}_{t-1}, \beta_t\mathbf{\Sigma})\mathcal{N}(\mathbf{x}_{t-1}; \sqrt{\bar{\alpha}_{t-1}}\tilde{\mathbf{x}}_0, (1 - \bar{\alpha}_{t-1}\mathbf{\Sigma}))}{\mathcal{N}(\mathbf{x}_t; \sqrt{\bar{\alpha}_t}\tilde{\mathbf{x}}_0, (1 - \bar{\alpha}_t)\mathbf{\Sigma})}$$

$$= (2\pi\tilde{\beta}_t)^{-\frac{d}{2}} \exp\left(-\frac{1}{2\tilde{\beta}_t} \left\| \mathbf{x}_{t-1} - \frac{\sqrt{\bar{\alpha}_{t-1}}\beta_t}{1 - \bar{\alpha}_t}\tilde{\mathbf{x}}_0 \right\|_{\mathbf{\Sigma}^{-1}}^2\right),$$

where $\|\mathbf{x}\|_{\mathbf{\Sigma}^{-1}}^2 = \mathbf{x}^T \mathbf{\Sigma}^{-1} \mathbf{x}$. Therefore,

$$q(\mathbf{x}_{t-1}|\mathbf{x}_t, \tilde{\mathbf{x}}_0) = \mathcal{N}\left(\mathbf{x}_{t-1}; \frac{\bar{\alpha}_{t-1}\beta}{1 - \bar{\alpha}_t}\tilde{\mathbf{x}}_0 + \frac{\sqrt{\alpha_t(1 - \bar{\alpha}_{t-1})}}{1 - \bar{\alpha}_t}\mathbf{x}_t, \tilde{\beta}_t\mathbf{\Sigma}\right)$$

Then, we calculate each term of the ELBO expansion. The first term is

$$\mathbb{E}_q \mathrm{KL}(q(\mathbf{x}_T|\tilde{\mathbf{x}}_0)\|p(\mathbf{x}_T)) = \frac{\bar{\alpha}_T}{2} \mathbb{E}_{\tilde{\mathbf{x}}_0} \|\tilde{\mathbf{x}}_0\|_{\mathbf{\Sigma}^{-1}}^2 - \frac{d}{2}(\bar{\alpha}_T + \log(1 - \bar{\alpha}_T)). \tag{10}$$

The second term is

$$\mathbb{E}_q \mathrm{KL}(q(\mathbf{x}_{t-1}|\mathbf{x}_t, \tilde{\mathbf{x}}_0)\|p_\theta(\mathbf{x}_{t-1}|\mathbf{x}_t)) = \frac{\beta_t}{2\alpha_t(1 - \bar{\alpha}_{t-1})} \mathbb{E}_{\tilde{\mathbf{x}}_0, \epsilon} \|\epsilon - \epsilon_\theta(\mathbf{x}_t, t)\|_{\mathbf{\Sigma}^{-1}}^2. \tag{11}$$

Finally, we have

$$\mathbb{E}_q \log p_\theta(\tilde{\mathbf{x}}_0|\mathbf{x}_1) = -\frac{1}{2}\log(2\pi\beta_1)^d \det(\mathbf{\Sigma}) - \frac{1}{2\alpha_1} \mathbb{E}_{\tilde{\mathbf{x}}_0, \epsilon} \|\epsilon - \epsilon_\theta(\mathbf{x}_1, 1)\|_{\mathbf{\Sigma}^{-1}}^2. \tag{12}$$

Combining Equation (10), (11) and (12) together, we can get the result in the Proposition.

## A.2 THEORETICAL BENEFITS OF PRIORGRAD

**Proposition 2** *Let* $L(\mu, \mathbf{\Sigma}, \mathbf{x}_0; \theta)$ *denote the* $-$*ELBO loss in Proposition 1. Suppose that* $\epsilon_\theta$ *is a linear function. Under the constraint that* $\det(\mathbf{\Sigma}) = \det(\mathbf{I})$*, we have* $\min_\theta L(\mu, \mathbf{\Sigma}, \mathbf{x}_0; \theta) \leq \min_\theta L(\mathbf{0}, \mathbf{I}, \mathbf{x}_0; \theta)$*.*

*Proof*: We use $L(\boldsymbol{\mu}, \boldsymbol{\Sigma}, \mathbf{x}_0; \theta)$ to denote -ELBO. According to Equation (10), (11) and (12), we have

$$L(\boldsymbol{\mu}, \boldsymbol{\Sigma}, \mathbf{x}_0; \theta) = \frac{\bar{\alpha}_T}{2}\mathbb{E}_{\mathbf{x}_0}\|\tilde{\mathbf{x}}_0\|^2_{\boldsymbol{\Sigma}^{-1}} + \frac{1}{2}\log\det(\boldsymbol{\Sigma}) + \sum_{t=1}^{T}\gamma_t\mathbb{E}_{\mathbf{x}_0,\boldsymbol{\epsilon}}\|\boldsymbol{\epsilon} - \boldsymbol{\epsilon}_\theta(\sqrt{\bar{\alpha}_t}\tilde{\mathbf{x}}_0 + \sqrt{1-\bar{\alpha}_t}\boldsymbol{\epsilon}, t)\|^2_{\boldsymbol{\Sigma}^{-1}}$$

$$+ \frac{d}{2}\log(2\pi\beta_1) - \frac{d}{2}(\bar{\alpha}_T + \log(1-\bar{\alpha}_T)),$$

where $\tilde{\mathbf{x}}_0 = \mathbf{x}_0 - \boldsymbol{\mu}$. We assume that $\boldsymbol{\epsilon}_\theta(\cdot)$ is a scalar function with parameter freedom $d$, i.e., $\boldsymbol{\epsilon}_\theta(\sqrt{\bar{\alpha}_t}\tilde{\mathbf{x}}_0 + \sqrt{1-\bar{\alpha}_t}\boldsymbol{\epsilon}, t) = \theta(\sqrt{\bar{\alpha}_t}\tilde{\mathbf{x}}_0 + \sqrt{1-\bar{\alpha}_t}\boldsymbol{\epsilon})$, where $\theta \in \mathbb{R}^{d\times d}$ with constraint $\tilde{\theta} = \mathbf{Q}\theta = diag(\tilde{\theta}_1, \cdots, \tilde{\theta}_d)$ and $\mathbf{Q}$ is the eigenmatrix for $\boldsymbol{\Sigma}^{-1}$, i.e., $\boldsymbol{\Sigma}^{-1} = \mathbf{Q}^T\tilde{\boldsymbol{\Sigma}}^{-1}\mathbf{Q}$ with $\tilde{\boldsymbol{\Sigma}}^{-1} = diag(\frac{1}{\sigma_1}, \cdots, \frac{1}{\sigma_d})$. For the term $\sum_{t=1}^{T}\gamma_t\mathbb{E}_{\mathbf{x}_0,\boldsymbol{\epsilon}}\|\boldsymbol{\epsilon} - \theta(\sqrt{\bar{\alpha}_t}\tilde{\mathbf{x}}_0 + \sqrt{1-\bar{\alpha}_t}\boldsymbol{\epsilon})\|^2_{\boldsymbol{\Sigma}^{-1}}$, we have

$$\sum_{t=1}^{T}\gamma_t\mathbb{E}_{\mathbf{x}_0,\boldsymbol{\epsilon}}\|\boldsymbol{\epsilon} - \theta(\sqrt{\bar{\alpha}_t}\tilde{\mathbf{x}}_0 + \sqrt{1-\bar{\alpha}_t}\boldsymbol{\epsilon})\|^2_{\boldsymbol{\Sigma}^{-1}}$$

$$= \sum_{t=1}^{T}\gamma_t\mathbb{E}_{\mathbf{x}_0,\boldsymbol{\epsilon}}\|Q(\boldsymbol{\epsilon} - \theta(\sqrt{\bar{\alpha}_t}\tilde{\mathbf{x}}_0 + \sqrt{1-\bar{\alpha}_t}\boldsymbol{\epsilon}))\|^2_{\tilde{\boldsymbol{\Sigma}}^{-1}}$$

$$= \sum_{j=1}^{d}\left(\frac{1}{\sigma_j}\sum_{t=1}^{T}\gamma_t\left(\sigma_j + (1-\bar{\alpha}_t)\tilde{\theta}_j^2\sigma_j + \bar{\alpha}_t\tilde{\theta}_j^2\sigma_j - 2\sigma_j\sqrt{1-\bar{\alpha}_t}\tilde{\theta}_j\right)\right)$$

$$= \sum_{j=1}^{d}\left(\frac{1}{\sigma_j}\sum_{t=1}^{T}\gamma_t\left(\sigma_j + \tilde{\theta}_j^2\sigma_j - 2\sigma_j\sqrt{1-\bar{\alpha}_t}\tilde{\theta}_j\right)\right)$$

$$= \sum_{j=1}^{d}\left(\sum_{t=1}^{T}\gamma_t\left(1 + \tilde{\theta}_j^2 - 2\sqrt{1-\bar{\alpha}_t}\tilde{\theta}_j\right)\right) \tag{13}$$

where the second equation is established by taking expectation of $\mathbf{x}_0$ and $\boldsymbol{\epsilon}$. The above equation will achieve its minimum at $\tilde{\theta}_j = \frac{\sum_{t=1}^{T}\gamma_t\sqrt{1-\bar{\alpha}_t}}{\sum_{t=1}^{T}\gamma_t}, \forall j$, and the minimum is $\sum_{j=1}^{d}\left(\sum_{t=1}^{T}\gamma_t - \frac{(\sum_{t=1}^{T}\gamma_t\sqrt{1-\bar{\alpha}_t})^2}{\sum_{t=1}^{T}\gamma_t}\right)$.

For $L(\mathbf{0}, \mathbf{I}, \mathbf{x}_0; \theta)$, we have

$$L(\mathbf{0}, \mathbf{I}, \mathbf{x}_0; \theta) = \frac{\bar{\alpha}_T}{2}\mathbb{E}_{x_0}\|\mathbf{x}_0\|^2 + \frac{1}{2}\log\det(\mathbf{I}) + \sum_{t=1}^{T}\gamma_t\mathbb{E}_{\mathbf{x}_0,\boldsymbol{\epsilon}}\|\boldsymbol{\epsilon} - \theta(\sqrt{\bar{\alpha}_t}\mathbf{x}_0 + \sqrt{1-\bar{\alpha}_t}\boldsymbol{\epsilon}, t)\|^2$$

$$+ \frac{d}{2}\log(2\pi\beta_1) - \frac{d}{2}(\bar{\alpha}_T + \log(1-\bar{\alpha}_T)),$$

Similarly, we can get the minimum of $\sum_{t=1}^{T}\gamma_t\mathbb{E}_{\mathbf{x}_0,\boldsymbol{\epsilon}}\|\boldsymbol{\epsilon}-\theta(\sqrt{\bar{\alpha}_t}\mathbf{x}_0+\sqrt{1-\bar{\alpha}_t}\boldsymbol{\epsilon}, t)\|^2$ with a diagonal $\theta$ is $\sum_{j=1}^{d}\left(\sum_{t=1}^{T}\gamma_t - \frac{(\sum_{t=1}\gamma_t\sqrt{1-\bar{\alpha}_t})^2}{\sum_{t=1}^{T}\gamma_t(1-\bar{\alpha}_t+\bar{\alpha}_t\sigma_j)}\right)$. Under the condition that $\det(\boldsymbol{\Sigma}) = \det(\mathbf{I})$, we have the minimum of $\sum_{j=1}^{d}\left(\sum_{t=1}^{T}\gamma_t - \frac{(\sum_{t=1}\gamma_t\sqrt{1-\bar{\alpha}_t})^2}{\sum_{t=1}^{T}\gamma_t(1-\bar{\alpha}_t+\bar{\alpha}_t\sigma_j)}\right)$ (over all possibilities of $(\sigma_1, \cdots, \sigma_d)$) will be $\sum_{j=1}^{d}\left(\sum_{t=1}^{T}\gamma_t - \frac{(\sum_{t=1}\gamma_t\sqrt{1-\bar{\alpha}_t})^2}{\sum_{t=1}^{T}\gamma_t}\right)$ (by solving the constrained minimization problem). Thus, we get the result in the Proposition.

**Remark:** From Equation (12), we have the second order derivative of $L(\boldsymbol{\mu}, \boldsymbol{\Sigma}, \mathbf{x}_0; \theta)$ with respect to $\theta_j$ equals $\sum_{t=1}^{T}\gamma_t$ for all $j = 1, \cdots, d$. Thus the condition number of the Hessian matrix equals 1. Similarly, the second order derivative of $L(\mathbf{0}, \mathbf{I}, \mathbf{x}_0; \theta)$ with respect to $\theta_j$ equals $\sum_{t=1}^{T}\gamma_t(1-\bar{\alpha}_t+\bar{\alpha}_t\sigma_j)$. Thus the Hessian matrix equals $c_1\mathbf{I} + c_2\boldsymbol{\Sigma}$ with $c_1 = \sum_{t=1}^{T}\gamma_t(1-\bar{\alpha}_t)$ and $c_2 = \sum_{t=1}^{T}\gamma_t\bar{\alpha}_t$, whose condition number is no less than 1.

### A.3 EXPLORATION ON THE CONDITIONAL INFORMATION SOURCE OF PRIORGRAD VOCODER

In this section, we provide further discussion regarding the source of the conditional information for constructing PriorGrad vocoder and empirical justification of selecting the frame-level spectral

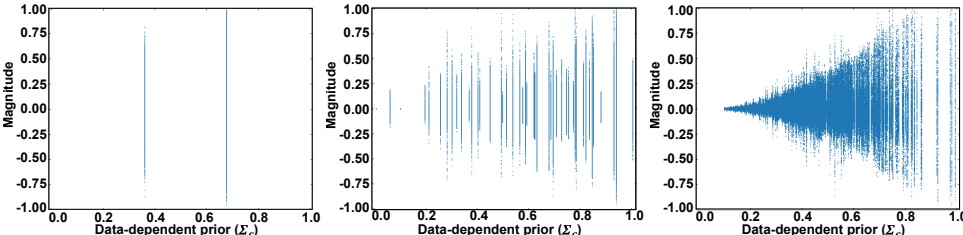

Figure 6: Scatter plots of waveform audio signals from the test set under different choices of the conditional information for PriorGrad vocoder. Left: V/UV label-based prior. Middle: Phoneme label-based prior. Right: Energy-based prior.

energy as the prior. Considering the waveform synthesis model as a vocoder, the main application of such model is a spectrogram inversion module of the text-to-speech pipeline. That is, the vocoder model is usually combined with the acoustic model as a front-end, where the acoustic model generates the mel-spectrogram based on text input. FastSpeech 2 (Ren et al., 2020) demonstrated a significant improvement in the quality of the acoustic model leveraged by carefully designed additional speech-related features for fine-grained supervision, such as voiced/unvoiced (V/UV) label obtained from F0 contours, or phoneme-to-frame alignment labels. We explored V/UV or phoneme labels as alternative sources of information for constructing the prior.

Figure 6 shows scatter plots of the waveform audio clip from the test set corresponding to the standard deviation labels assigned by $\Sigma_c$ acquired from the training set. Because the V/UV label is binary, we can only assign two different prior variances to the waveform distribution, which is an overly coarse assumption of the data distribution. Phoneme-level prior can offer more fine-grained approximate prior. However, we found that the variance statistics acquired from the training set is misaligned with the unobserved waveform, where the actual variance can be significantly different to the target prior, leading to an inconsistent result. The frame-level spectral energy for PriorGrad exhibited the best alignment of the label and the unobserved waveform data and demonstrated consistent results in quality.

## A.4  DESCRIPTION OF OBJECTIVE METRICS FOR PRIORGRAD VOCODER

In this section, we describe the objective metrics we collected to evaluate the performance of PriorGrad vocoder.

**Log-mel spectrogram mean absolute error (LS-MAE)**  This is a spectral regression error between the log-mel spectrogram computed between the synthesized waveform and the ground-truth. We used the same STFT function for computing the mel-spectrogram as the conditional input of DiffWave and PriorGrad.

**Multi-resolution STFT error (MR-STFT)**  This measures the spectral distance across multiple resolutions of STFT. MR-STFT is widely adopted as a training objective of recent neural vocoders because using multiple resolution windows can capture the time-frequency distributions of the realistic speech signal. We used the resolution hyperparameter proposed in Parallel WaveGAN (Yamamoto et al., 2020), which is implemented in an open-source library[4].

**Mel-cepstral distortion (MCD)**  MCD is a widely adopted speech metric (Kubichek, 1993) which measures the distance between the mel cepstra. We used an open-source implementation with default parameters[5].

**$F_0$ root mean square error ($F_0$ RMSE)**  This measures the accuracy (measured by RMSE) of the fundamental frequency ($F_0$) which is an approximate frequency of the (quasi-)periodic structure of

---

[4]https://github.com/csteinmetz1/auraloss
[5]https://github.com/MattShannon/mcd

Table 5: Sampling noise schedule used for PriorGrad acoustic model experiments obtained by a grid search method (Chen et al., 2021).

| Method | $T_{infer}$ | Sampling Noise Schedule |
|---|---|---|
| Baseline | 2 | [0.8, 0.9] |
| | 6 | [0.0006, 0.003, 0.01, 0.07, 0.8, 0.9] |
| | 12 | [0.0002, 0.0005, 0.002, 0.008, 0.03, 0.04, 0.06, 0.07, 0.5, 0.7, 0.8, 0.9] |
| PriorGrad | 2 | [0.3, 0.9] |
| | 6 | [0.0001, 0.008, 0.01 0.05, 0.7, 0.9] |
| | 12 | [0.0002, 0.0007, 0.004, 0.009, 0.01, 0.02, 0.06, 0.08, 0.1, 0.3, 0.5, 0.9] |

voiced speech signals. We used the Saw-tooth Waveform Inspired Pitch Estimation (SWIPE) (Camacho & Harris, 2008) with a hop size of 128 to obtain $F_0$ using an open-source implementation[6].

**Debiased Sinkhorn divergence** This is a positive definite approximation of optimal transport (*Wasserstein*) distance between two data distributions (Feydy et al., 2019). This metric can be used as a mathematically grounded assessment of the informative prior, which is evidenced by a recent study that the diffusion process coincides with the optimal transport map (Khrulkov & Oseledets, 2022). We used the open-source library [7] with default parameters. Using DiffWave and PriorGrad with $T_{infer} = 6$, We calculated the test set average of the Sinkhorn divergence using Monte Carlo estimate with 100 samples per test data point. We denote $S(x_T, x_0)$ as the distance between the prior and the real data, and $S(\tilde{x}_0, x_0)$ as the distance between the generated samples and the real data.

## A.5 ADDITIONAL DETAILS OF PRIORGRAD ACOUSTIC MODEL

In this section, we describe additional details of the experimental design of the PriorGrad acoustic model. The feed-forward Transformer-based phoneme encoder has 11.5M parameters trained with the Adam optimizer with the learning rate schedule identically described in Ren et al. (2020). The diffusion decoder is simultaneously trained with the same training configurations.

For a fair comparative study between different diffusion models, we searched for the best performing baseline model with $\mathcal{N}(\mathbf{0}, \mathbf{I})$ first, then applied PriorGrad to this baseline. We applied $T = 400$ for training the diffusion decoder with a linearly spaced beta schedule ranging from $1 \times 10^{-4}$ to $5 \times 10^{-2}$. We found that $1 \times 10^{-4}$ to $2 \times 10^{-2}$ used for waveform domain in Kong et al. (2021) performed poorly for the baseline, where the model failed to capture the high-frequency details of the mel-spectrogram. This indicates that the optimal training noise schedule can be different, depending on the data domain (Kong & Ping, 2021). On the contrary, PriorGrad's performance was similar under different choices of training noise schedules, suggesting that PriorGrad also features robustness regarding designing the noise schedules for model training.

We found that the fast reverse noise schedule with $T_{infer} = 6$ described in Kong et al. (2021) performed poorly for the baseline model with $\mathcal{N}(\mathbf{0}, \mathbf{I})$. Thus, we applied grid search over the $T_{infer} = 6$ reverse schedule for each model, which is a similar approach to that in Chen et al. (2021). This enabled a fair comparative study of each model configuration when using the fast noise schedule for sampling. We applied a fine-grained grid search over every possible combination of the monotonically increasing betas based on the L1 loss between the model prediction and the target from the validation set. We applied the following range for the grid search: $\{1, 2, ..., 8, 9\} \times \{10^{-1}, 10^{-1}\}$ for $T_{infer} = 2$, $\{1, 2, ..., 8, 9\} \times \{10^{-4}, 10^{-3}, 10^{-2}, 10^{-2}, 10^{-1}, 10^{-1}\}$ for $T_{infer} = 6$, and $\{1, 2, ..., 8, 9\} \times \{10^{-4}, 10^{-4}, 10^{-3}, 10^{-3}, 10^{-2}, 10^{-2}, 10^{-2}, 10^{-2}, 10^{-1}, 10^{-1}, 10^{-1}, 10^{-1}\}$ for $T_{infer} = 12$.

We found that setting relatively high values of beta at the last steps was important for the quality of the baseline model, whereas PriorGrad was significantly more robust to the choice of the noise schedule. Table 5 shows the optimal beta schedules for each model configuration from the grid search method.

---

[6]https://github.com/r9y9/pysptk
[7]https://www.kernel-operations.io/geomloss/

Table 6: Additional MOS results of acoustic models including an alternative method with jointly trainable estimation of the diffusion prior. We used a pre-trained Parallel WaveGAN (Yamamoto et al., 2020) for the vocoder.

| Method | Parameters (Decoder) | Training Steps | |
|---|---|---|---|
| | | 60K | 300K |
| Baseline | 10M | $3.84 \pm 0.10$ | $3.91 \pm 0.09$ |
| PriorGrad | 10M | $4.04 \pm 0.07$ | $\mathbf{4.09 \pm 0.08}$ |
| Trainable $\mathcal{N}(\boldsymbol{\mu}_\theta, \boldsymbol{\Sigma}_\theta)$ | 10M | FAIL | $3.32 \pm 0.12$ |

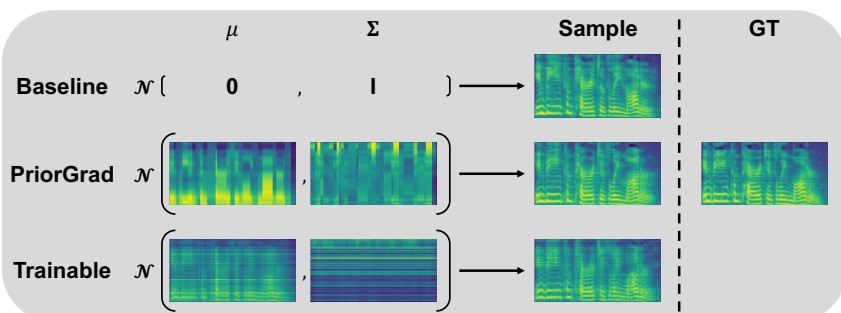

Figure 7: Visualized example of PriorGrad acoustic model. Top: Baseline model generates data from the standard Gaussian, leading to higher level of noise and slow training. Middle: PriorGrad model generates data from the data-dependent non-standard Gaussian that improves the quality and accelerates model training. Bottom: Alternative model with jointly trainable diffusion prior, where the estimation is noisy and quality is worse. Right: ground-truth mel-spectrogram.

## A.6    COMPARISON TO TRAINABLE DIFFUSION PRIOR

In this section, we present additional results regarding the challenges of formulating a jointly trainable estimation of the diffusion prior. Based on the improvements achieved by PriorGrad, it is natural to suppose that the jointly trainable diffusion prior distribution with additional parameterization might perform similar or better than the data-dependent prior. In this section, we explored a feasibility of the diffusion prior with trainable parameters. For acoustic model, we explored two setups: 1. replacing the fixed data-dependent prior with the estimated $\mathcal{N}(\boldsymbol{\mu}_\theta, \boldsymbol{\Sigma}_\theta)$ from the projection layer of the phoneme encoder, 2. parameterizing $\mathcal{N}(\boldsymbol{\mu}_\theta, \boldsymbol{\Sigma}_\theta)$ with an independently defined convolutional layers with the data-dependent $(\boldsymbol{\mu}, \boldsymbol{\Sigma})$ we applied to PriorGrad as an input to acquire a refined prior. For the vocoder model, instead of the spectral energy as the proxy of the waveform variance, we explored an independently defined diffusion prior estimator by using the smaller-scale convolutional network with the mel-spectrogram as the input.

We found that all approaches failed to jointly estimate or refine the informative prior distribution. For the acoustic model, the estimated mean was significantly noisier than our data-dependent one, and the estimated variance is collapsed to a single point which is uninformative. These models performed significantly worse than the baseline method with the fixed $\mathcal{N}(\mathbf{0}, \mathbf{I})$, as measured by Table 6. We visualize the data-dependent prior compared to the estimated version in Figure 7. For the vocoder model, the diffusion prior estimator resulted in a divergence of the estimated mean and collapse of the estimated variance to zero, leading to the training failure. We do not draw the conclusive claim that the jointly trainable diffusion prior is not possible. However, the results suggest that the fixed data-dependent diffusion prior with PriorGrad is an effective and easy-to-use method to improve the efficiency of the diffusion-based generative model without introducing additional complexities to the network.

Table 7: Expanded vocoder model results compared to previous work with 95% confidence intervals. †: pretrained weights obtained from the open-source repository with different train/test split.

| Method | $T_{infer}$ | MOS | RTF | Parameters |
|---|---|---|---|---|
| GT | - | $4.60 \pm 0.05$ | - | - |
| DiffWave / PriorGrad | 6 | $4.10 \pm 0.08$ / $4.20 \pm 0.08$ | 0.1388 | 2.62M |
|  | 12 | $4.15 \pm 0.08$ / $4.29 \pm 0.08$ | 0.2780 |  |
|  | 50 | $4.19 \pm 0.07$ / $\mathbf{4.33 \pm 0.07}$ | 1.1520 |  |
| WaveGlow † | - | $4.09 \pm 0.08$ | 0.0780 | 87.9M |
| WaveFlow † | - | $4.01 \pm 0.09$ | 0.1759 | 22.3M |
| HiFi-GAN (V1) † | - | $\mathbf{4.44 \pm 0.05}$ | $\mathbf{0.0068}$ | 14.0M |

Table 8: Expanded acoustic model results compared to previous work with 95% confidence intervals. We used a pretrained HiFi-GAN (V1) (Kong et al., 2020) for the vocoder. †: pretrained weights obtained from the open-source repository with different train/test split.

| Method | $T_{infer}$ | MOS | RTF | Parameters | |
|---|---|---|---|---|---|
|  |  |  |  | Encoder | Decoder |
| GT | - | $4.65 \pm 0.05$ | - | - | - |
| GT (Vocoder) | - | $4.50 \pm 0.06$ | - | - | - |
| Baseline / PriorGrad | 2 | $2.80 \pm 0.17$ / $4.25 \pm 0.08$ | $\mathbf{0.0069}$ | 11.5M | 3.5M |
|  | 6 | $3.67 \pm 0.12$ / $4.29 \pm 0.07$ | 0.0113 |  |  |
|  | 12 | $4.14 \pm 0.08$ / $\mathbf{4.39 \pm 0.08}$ | 0.0176 |  |  |
| Grad-TTS† | 2 | $3.43 \pm 0.15$ | 0.0090 | 7.2M | 7.6M |
|  | 10 | $\mathbf{4.38 \pm 0.05}$ | 0.0308 |  |  |
| FastSpeech 2 | - | $4.19 \pm 0.08$ | $\mathbf{0.0040}$ | 11.5M | 11.5M |
| Glow-TTS† | - | $4.23 \pm 0.08$ | 0.0081 | 7.2M | 21.4M |

## A.7 COMPARISON TO STATE-OF-THE-ART

In this section, we present additional results of PriorGrad by comparing to recent state-of-the-art speech synthesis models and show that PriorGrad is competitive with or sometimes outperforms the previous models. We provide a detailed analysis with a varying number of inference denoising steps ($T_{infer}$), along with their speed measured by a real-time factor (RTF) on the NVIDIA A40 GPU, and the model capacity measured by the number of parameters.

**Vocoder comparison**  Table 7 provides an expanded MOS result of the fully converged PriorGrad vocoder with the 1M training steps. For the fast inference noise schedule, we used $T_{infer} = 6$ defined as [0.0001, 0.001, 0.01, 0.05, 0.2, 0.5] from DiffWave$_{BASE}$ without modification, and used $T_{infer} = 12$ defined as [0.0001, 0.0005, 0.0008, 0.001, 0.005, 0.008, 0.01, 0.05, 0.08, 0.1, 0.2, 0.5]. For the full $T_{infer} = 50$, PriorGrad significantly outperformed the baseline DiffWave. Furthermore, PriorGrad achieved an even larger performance gap with a significantly reduced network capacity, compared to the flow-based vocoders such as WaveGlow (Prenger et al., 2019) and WaveFlow (Ping et al., 2020). This suggests that PriorGrad is a competitive likelihood-based neural vocoder, and enables a step closer to the current state-of-the-art GAN-based model (HiFi-GAN, Kong et al. (2020)). The speed is close to real-time (RTF being close to 1), but it is noticeably slower than other approaches.

For the fast inference with $T_{infer} = 6$ denoising steps, PriorGrad was able to match the performance to the baseline DiffWave model with $T_{infer} = 50$ while still outperforming the flow-based vocoders with comparable RTF. This further showcase the efficiency of PriorGrad, where having access to the informative non-standard Gaussian prior accelerates the inference of the diffusion-based model.

**Acoustic model comparison**  We additionally trained our baseline diffusion-based acoustic model and the enhanced model with PriorGrad which are compatible with the pretrained HiFi-GAN (Kong et al., 2020) vocoder. This enabled a fair assessment of PriorGrad compared to recent state-of-the-art acoustic models. We trained our acoustic model for the full convergence with 1M training steps. We found that the small PriorGrad with 3.5M decoder parameters performed almost identical to the high-capacity (10M) model. Therefore, we chose the small decoder model as our final choice for comparison.

Table 8 provides an expanded MOS result of PriorGrad acoustic model with varying $T_{infer}$ and comparison to the recent representative acoustic model from different categories: Feed-forward (FastSpeech 2, Ren et al. (2020)), Flow-based (Glow-TTS, Kim et al. (2020)), and the concurrent diffusion-based model (Grad-TTS, Popov et al. (2021)). The results show that PriorGrad acoustic model is competitive to the state-of-the-art models. PriorGrad with $T_{infer} = 12$ provided an identical quality to the state-of-the-art Grad-TTS with $T_{infer} = 10$. This is achieved with an approximately 1.75 times faster RTF and sets PriorGrad a new state-of-the-art acoustic model.

Armed with the informative non-standard Gaussian prior, PriorGrad is robust to the extremely small number of inference steps with $T_{infer} = 2$. The fast PriorGrad matches the performance with the state-of-the-art flow-based acoustic model, Glow-TTS, with fewer parameters. The RTF becomes close to the feed-forward FastSpeech 2 as well as scoring higher MOS. By contrast, the baseline acoustic model with $\mathcal{N}(\mathbf{0}, \mathbf{I})$ suffered a significant degradation for smaller $T_{infer}$. This result further highlights the efficiency of PriorGrad, where it is robust to the reduced number of denoising steps and enables a significantly accelerated inference.

## A.8 MOS EVALUATION DETAILS

We conducted a standard and widely adopted protocol for the MOS evaluation using Amazon Mechanical Turk. We constructed 20 randomly selected clips from the test set, and applied a total of 450 evaluations for each model with the 5-scale scoring on audio naturalness. To encompass the diverse set of listeners, a single listener can only evaluate the three randomly exposed samples. Therefore, there are 150 unique listeners for each model. For reliability, only skilled listeners with the evaluation approval rate higher than 98% and the number of previously approved evaluations higher than 100 are qualified to conduct the test. The listeners were rewarded $0.2 for each evaluation. Approximately $130 were spent as compensation for each set of the MOS test.

## A.9 AUDIO SAMPLES DEMO PAGE

We provide a demo page of the audio samples used in this study: `https://speechresearch.github.io/priorgrad/`

