# OpenReview forum: "PriorGrad: Improving Conditional Denoising Diffusion Models with Data-Dependent Adaptive Prior"
_ICLR.cc/2022/Conference — ICLR 2022 Poster_

### Official Review · Reviewer_518v · 2021-10-20

**Correctness:** 3
**Technical Novelty And Significance:** 2
**Empirical Novelty And Significance:** 3
**Recommendation:** 6
**Confidence:** 4

**Main Review:**

Strengths:
- Time-wise and instance-based means and variances provide additional information that models are able to exploit, yielding improved results in terms of convergence speed and generation quality.
- Results are reported for both objective and subjective assessments. Baselines are clear and appropriate.
- As far as I can judge, I think the paper is technically correct.
- Well-written paper, except from some minor typos (see below).

Weaknesses:
- Obvious adaptation from standard to non-standard Gaussian prior.
- Potential lack of generality to other conditional audio tasks or other domains such as image.
- Non-trivial and case-specific methodology to compute means and variances for the non-standard Gaussian priors.
- I don't think one of the three main claims is well supported by the experiments (see below).

Extended review:
1. I have the impression that the title, parts of the text, and even the abstract, try to encompass a general application of the methodology. However, results are only provided for two specific speech processing tasks. The title, for instance, does not even mention speech, and the abstract mentions audio when only speech is considered. While in principle the proposed methodology could be applied to other tasks and domains, the benefit of it is not shown for those.
1. The procedure to compute instance- and time-specific means and variances for the prior seems non-straightforward and highly dependent on the type of conditioning signal, further complicating the generalization of the approach. The authors even state that with other conditioning signals for the considered task (V/U segments and phoneme-level statistics) they did not obtain any improvements. This also raises some doubts beyond generalization. In particular, regarding how hard is to adapt the approach to other tasks/domains or even conditioning signals, or how limited is the proposed idea. Even in the acoustic modeling setting the approach needs to rely on mel band energies (leaving the reader suspicious as if this is the only information that works).
1. There is some unclear formulation of the approach. Expanding the last point to focus more on the considered case: How is frame-level energy normalized? Which is the exact formula? How do different procedures affect final performance? Why there should be a range of $(0,1]$ (later the paper mentions it is not really $(0,1]$ but clipped to some empirically-chosen value...)?
1. There is at least concurrent work (https://arxiv.org/abs/2110.05948) investigating the effect of using other priors beyond the standard Gaussian. Perhaps contribution 1 could be softened in this regard.
1. No details on how MOS subjective tests were performed. This can be a serious issue.
1. In my opinion, the "tolerance to a reduction in network capacity" claim is not supported by the results. First of all, it is debatable whether reducing the width of convolutional layers is sound or meaningful in this setting (Sec 4.3, "parameter efficiency"). But, more importantly, the relative change in MOS in Table 2 does not support the claim (3.9/4.06=0.961 which is extremely close to 4.02/4.12=0.975; that is, both DiffWave and PriorGrad present almost the same relative reduction in performance (4% vs 3%). This complexity-leverage claim again does not hold with results in Table 4 when we think of relative terms.

Minor considerations:
- Talking about "inefficiency" in conditional speech synthesis when for instance WaveGrad shows that with only 6 steps one can achieve almost perfect accuracy is, at least, misleading.
- "Given the above investigation" --> Given the previous reasoning?
- "our method can provide a better trajectory..." --> Claim not shown either theoretically or empirically.
- In Algorithm 2, it is perhaps unclear that $\sigma_t$ corresponds to a noise source or that it includes it (the authors use $\epsilon$ for Algorithm 1).
- I think Figs 3 and 5 are unnecessary.
- "by comparing the baseline" --> by comparing to the baseline
- References Jeong et al and Kawar et al have no venue or arxiv or paper link.



**Summary Of The Paper:**

The paper proposes to use a data-dependent, adaptive prior for the noise used in conditional DDPMs. In particular, it proposes to move from the standard to the non-standard Gaussian prior. For that, the ELBO loss is reformulated by taking into account adaptive means and variances and the sampling procedure is also adapted. Statistics (directly or indirectly derived) from conditioning signals are used to compute time-wise and instance-based means/standard deviations. An empirical evaluation is conducted on two conditional audio generation tasks: vocoding and acoustic modeling, showing some favorable results.

**Summary Of The Review:**

The paper is technically correct and shows some improvements in the empirical evaluation. However, I think that moving from standard to non-standard Gaussian priors is rather obvious. In addition, I am concerned about the generality of the approach, as it only focuses on two specific speech generation tasks and deriving appropriate means and variances from certain conditioning signals can be problematic or may not yield to an improvement. One of the three main claims is not well supported.

[Update: After authors' rebuttal I'm changing my score from 5 to 6]

---

> ### Author Response · Authors · 2021-11-23
> **Response to reviewer 518v (part 1/2)**
>
> Thank you for the constructive feedback. We provide a point-to-point response as below:
>
> **Q1: I think that moving from standard to non-standard Gaussian priors is rather obvious.**
>
> A1: We understand that the idea of expanding the current DDPM with the standard Gaussian to the non-standard one might look rather obvious at first sight. However, we would like to point out that correctly formulating the expansion to the non-standard Gaussian requires non-trivial derivations of the objective, as provided by the proof in the appendix. For example, applying an inverse of the data-dependent variance to the ELBO and shifting the mean to the data point for training are accurate realizations of the non-standard Gaussian expansion for the DDPM training, which can only be achieved from the rigorous mathematical derivation.
>
> **Q2: I am concerned about the generality of the approach: Non-trivial and case-specific methodology to compute means and variances for the non-standard Gaussian priors. Potential lack of generality to other conditional audio tasks or other domains such as image.**
>
> A2: We acknowledge that the realization of PriorGrad we presented is a task-specific endeavor. Indeed, we have discussed the current limitation of this work in the conclusion section in the manuscript in a transparent way, to foster the future study towards a generalized way to realize PriorGrad. While PriorGrad significantly improved the efficiency and quality of the speech synthesis models, the method may be more challenging depending on the granularity and resolution of the conditional information. What we claimed in this work is that if we have access to the approximate prior from the conditional information, leveraging the proxy greatly improves efficiency and quality. We have discussed future avenues where PriorGrad can potentially be applied on image domains (beyond the text-conditional spectrogram image generation in this work), such as using patch-level statistics for the image super-resolution, or the conditional variance for the depth-conditional image generation.
>
> With that said, we understand that the tone of the current manuscript might look over-encompassing the generalizability of the work. Reflecting on the feedback, we will tone down the writing regarding the scope of the work by explicitly stating that the study focuses on the speech synthesis domain to the manuscript wherever appropriate, including the abstract.
>
> **Q3: How is frame-level energy normalized? Which is the exact formula? How do different procedures affect final performance? Why there should be a range of (0,1] (later the paper mentions it is not really (0,1] but clipped to some empirically-chosen value...)?**
>
> A3: The frame-level spectral energy (defined by the roots of sum of exponential of the mel-spectrogram band, $e_i=\sqrt{\sum_{j=1}^{B}\exp{c_{ij}}}$, where $B$ is the number of band of the mel-spectrogram frame $c_i$), is normalized to (0, 1] by computing the minimum and maximum value of the spectral energy of the training dataset. The choice of normalization with a range to (0, 1] is to ensure that we cast the computed statistics to the standard deviation of non-standard Gaussian to have a range of (0, 1]. We can also design other ranges of the standard deviation, such as setting the maximum higher than 1. However, our preliminary study indicated that setting the value higher than 1 was worse in performance, resulting in setting the maximum to 1, which is a globally applied value in the baseline model.
>
> The clipping of minimum standard deviation is an empirical choice to stabilize model training. In our modified ELBO in Prop. 1, there is an inverse of the diagonal variance matrix. If there exists a too-small value (say, zero), its inverse could raise training instabilities (for example, a zero value will cause NaN which leads to training failure). Clipping the minimum of the standard deviation is also previously studied in the waveform synthesis literature (ClariNet, Ping et al., ICLR 2019), which ensures the training stability. The clipping threshold is agnostic to any speech corpora because it serves as a failsafe method to stabilize the model training for too-low variance.

---

> > ### Author Response · Authors · 2021-11-23
> > **Response to reviewer 518v (part 2/2)**
> >
> > **Q4: There is at least concurrent work (https://arxiv.org/abs/2110.05948) investigating the effect of using other priors beyond the standard Gaussian. Perhaps contribution 1 could be softened in this regard.**
> >
> > A4: Thank you for pointing out the concurrent work which studied other types of prior noise beyond the standard Gaussian. We will soften the contribution statement and add it to the related work.
> >
> > **Q5: No details on how MOS subjective tests were performed.**
> >
> > A5: We conducted a standard and widely adopted protocol for the MOS evaluation using Amazon mechanical turk (AMT): we constructed 20 randomly selected clips from the test set, and applied a total of 450 evaluations for each model with the 5-scale scoring on audio naturalness. To encompass the diverse set of evaluators, a single listener can only evaluate the three randomly exposed subsets. Therefore, there are 150 unique evaluators for each model. For reliability, only skilled listeners with an evaluation approval rate higher than 98% and the number of previous AMT evaluations higher than 100 are qualified to conduct the test. The listeners were rewarded 0.2 dollars for each evaluation. Approximately 130 dollars was spent as compensation for each set of the MOS test. We have added the aforementioned detail in appendix A.7.
> >
> > **Q6: the "tolerance to a reduction in network capacity" claim is not supported by the results. it is debatable whether reducing the width of convolutional layers is sound or meaningful.**
> >
> > A6: Thank you for raising the concern about the parameter efficiency claim, such as considering the relative MOS degradation on a linear scale. In defense of our parameter efficiency claim, we think that it is also debatable whether the subjective quality assessment score can be assessed numerically on a linear scale when assessing the relative performance. Due to its subjectiveness, multiple papers report different scores even for the same model. MOS is the go-to metric to show the relative rankings of the considered models, but it is hard to numerically analyze the score itself. It can also have many different views on which subjective or objective metrics we should use for the assessment. Regarding the way to control the model size, there can be many ways to restrict the network capacity. We believe that reducing the width can be regarded as one of the valid ways to reduce the expressiveness of the neural network.
> >
> > Our focus on parameter efficiency is from observing that our method, under restricted network capacity, can get close to or matches the high-capacity baseline model in relative rankings. This indicates that PriorGrad gets similar performance with fewer parameters. Another view on parameter efficiency is that our method performs better than baseline under the same network size, which indicates that the network can learn its defined weights more efficiently by leveraging the advanced methods we provide to the model. Nevertheless, we acknowledge that there can exist many different views on assessing parameter efficiency. We will consider an alternative way to better demonstrate the small model result.
> >
> > **Remarks**
> > We would like to note that we allocated our available resources preparing expanded experimental results of the presented speech synthesis models, reflecting the feedback of other reviewers. We hope the reviewer may further assess the significantly enhanced experimental results and our contribution, including model comparison with varying inference steps and previous SOTA model comparison in the updated manuscript. For example, the varying inference steps experiment provides empirical support of our “better diffusion trajectory” claim, where the baseline model has more degradation, whereas PriorGrad is robust. We believe that the additional results strengthened our contribution, such as achieving the new state-of-the-art acoustic model. We will also fix the text suggested in the minor consideration parts. Thank you again for the constructive feedback.

---

> > > ### Comment · Reviewer_518v · 2021-11-26
> > > **Thanks**
> > >
> > > Thank you for your answers and additions to the manuscript. I overall think the paper improved a bit and therefore I am raising my score from 5 to 6.

---

### Official Review · Reviewer_Wdw2 · 2021-10-25

**Correctness:** 3
**Technical Novelty And Significance:** 3
**Empirical Novelty And Significance:** 3
**Recommendation:** 6
**Confidence:** 4

**Main Review:**

This paper is well written and the key idea is described in a simple way. The main motivation stems from the question whether it is possible to induce diffusion probabilistic models (DDPM) to use a more informative prior, rather than the typical isotropic unit Gaussian, *without* additional computational or parameter complexity.

As a side comment, I would like to clarify that while it is true that the DDPM as formulated in this work bears no overheads, the task of estimating mean and covariances of the data (especially for "complex" data such as the one used in the experiments) comes with its own cost and design decisions, which should not underestimated.

The background section is typical for a DDPM paper (using the discrete-time Markov chain formulation). Although this is nitpicking, I think some clarifications are in order: the ELBO from eq (3), which is derived as in Ho et. al. 2020, is studied as three separate components, and the focus is on the second one, which is labeled $L_{t-1}$ in Ho et al. 2020. Then, eq (6) is not the ELBO, but only the $L_{t-1}$ term.
Also, since this work argues for taking into account an informative prior where $\mu$ and $\Sigma$ are instrumental, I would argue that the constant $C$ that appears in eq (6) should be developed, as there are terms that depend on it. This could help the reader gain a better understanding of the proposed method (and the limitations of proposition 1 and 2).

The presented method is discussed in an intuitive manner at first, using Fig 2. Nevertheless, after some thoughts, I think this figure can be misleading (even if it is just an illustration). In a two-dimensional toy example, $p(\mathbf{x}_T) = \mathcal{N}(\mu, \Sigma)$ could be "closer" to the true data distribution, actually, on top of it. This is not necessarily true for high-dimensional spaces though.
More to the point, since you "simulate" the transition from $\mathbf{x}_0$ to $\mathbf{x}_T$ in the forward process for a finite $T$, then the process does not end up in $ \mathcal{N}(0,\mathbf{I})$ (for the un-informative baseline) nor in $ \mathcal{N}(\mu, \Sigma)$ (for the data-dependent prior).
The backward process starts from the prior, and again, since it is "simulated" for a finite amount of time $T$, it will not reach $\mathbf{x}_0$.
This kind of illustration could be helpful to better understand the claim of proposition 2.

Also, note that in proposition 1, the quantity under study is not the ELBO, but $L_{t-1}$. This is also where the constant $C$ is "hiding" a dependence on $\mu$ and $\Sigma$.

Proposition 2 states that the use of a more informative prior is better in terms of a tighter bound. This seems intuitive at first, as there are more degrees of freedom than using zero mean and unit covariance. However, the proposition holds when the function that learns the denoising effect of the backward process is linear, which is not true in practice. Do you have any intuition if the proposition holds also in this more general case? Are there any other arguments that can be used to show that indeed the ELBO is smaller?

The observation that the Hessian of the $L_{t-1}$ term is better conditioned when using an informative prior w.r.t. the case of a unit Gaussian prior is also based on the assumption of linearity of the denoising function. However, results in Fig 4 do not show that the rate of convergence is improved in a practical endeavor. There, we see that the considered error metric is better for the informative prior case, but the rates of the two curves appear to be similar, just shifted. Do you have any comments on this effect?

Although the application domain used to evaluate the proposed method falls outside my competence, I appreciate the efforts and the timeliness of the architectures and design choices made in the execution of the experiments, which I find somehow compelling to corroborate that the paper presents a valid idea. That being said, the evaluation only compares the non-informative prior baseline against the informative variant. Wouldn't it be interesting to compare additional methods from the state of the art, which the authors appear to know very well? I particularly liked the idea of studying "parameter efficiency", and I wonder whether the efficiency of an informative prior carries on when comparing to different approaches.

To conclude my comments, I think this paper introduces a relevant idea which shows potential in applications. I am less excited about the methodological sections of this paper, which are not strong and precise enough to be convincing.


**Summary Of The Paper:**

This work builds on denoising diffusion probabilistic models (DDPM), and argues for to modify the forward and backward diffusion processes such that instead of using an uninformative prior $p(\mathbf{x}_T) = \mathcal{N}(0,\mathbf{I})$, they use a data-dependent prior $p(\mathbf{x}_T) = \mathcal{N}(\mu, \Sigma)$, where the mean and covariance are derived from training data, in a "pre-processing" step.

The authors show that, under restrictive conditions, the ELBO obtained with the proposed prior is smaller than that obtained with the uninformative prior. Under similar restrictive conditions, the authors argue that convergence rate of the parameter optimization is better conditioned, and as a consequence faster.

This work illustrates the benefits of the proposed method using two applications, one to a vocoder, and one to an acoustic model. Experiments rely on several very recent related work, especially concerning technical details of the architectures, and data pre-processing choices. Overall, two methods are compared, a baseline DDPM and the proposed DDPM that uses training data to build a better prior.

**Summary Of The Review:**

Interesting and relevant idea, to improve the behavior of diffusion models. Experimental results indicate the idea has a positive effect on metrics used in specific application domains, including training performance and parameter efficiency.
However, the theoretical justification of the proposed method is somehow weak, and does not provide sufficiently compelling arguments on the observed behavior of the proposed method in practice.


*** Post-rebuttal / discussions remarks ***

Thanks for the useful discussions and for the additional work to improve the paper. I have increased my ranking for the paper from 5 to 6.
The main reason why I didn't rank the paper higher is that the theoretical contribution of this work, despite being very promising, is still not mature and sufficiently mathematically grounded.

---

> ### Author Response · Authors · 2021-11-23
> **Response to Reviewer Wdw2**
>
> Thank you for the constructive feedback. We provide a point-to-point response as below:
>
> **Q1: eq (6) is not the ELBO, but only the Lt−1 term in the original DDPM paper. When μ and Σ are instrumental, the constant C that appears in eq (6) should be developed, as there are terms that depend on it. In proposition 1, the quantity under study is not the ELBO, but Lt−1. This is also where the constant C is "hiding" a dependence on μ and  Σ.**
>
> A1: Eq. (6) shows the ELBO as an objective of $\theta$, for which $C$ is a constant. Note the chosen $\mu$ and $\Sigma$ are fixed and not optimized in training the model, in order to serve as an input of information. The dependency on $\mu$ and $\Sigma$ is considered in Proposition 1, and we’ve made the dependency of $C$ on them explicit in the updated version.
>
> According to our proof for Proposition 1 in Appendix A.1:
> $$C = \frac{\bar{\alpha}_T}{2} E|\tilde{x}_0|_\Sigma^{-1} + \frac{1}{2}log (2\pi\beta_1)^d det(\Sigma)-\frac{d}{2}(\bar{\alpha_T}+log(1-\bar{\alpha}_T)). $$
>
>
> Only the term $\frac{\bar{\alpha}_T}{2} E|\tilde{x}_0|_\Sigma^{-1} + \frac{1}{2}log (2\pi\beta_1)^d det(\Sigma)-\frac{d}{2}(\bar{\alpha_T})$ relates with $(\mu,\Sigma)$.
>
> Solving the optimization problem: $\frac{\bar{\alpha}_T}{2} E|\tilde{x}_0|_\Sigma^{-1}$ , s.t. $det(\Sigma)$ is fixed, which is equivalent to
>
> $\sum_{j=1}^dx_j^2/\sigma_j$, s.t., we have $\Sigma=Cov(x_0)$.
>
> Therefore, setting the covariance matrix to be the covariance matrix of the input can make the term $C$ achieve the minimum.
>
>
> **Q2: The proposition 2 holds when the function that learns the denoising effect of the backward process is linear, which is not true in practice. Do you have any intuition if the proposition holds also in this more general case? Are there any other arguments that can be used to show that indeed the ELBO is smaller?**
>
> A2:  1) As we have explained in A1, the term $C$ in the ELBO will be decreased by the prior $N(\mu, \Sigma)$, which does not rely on the backward process.  2) For a non-linear function $\epsilon_{\theta}(x_t)$, such as neural network, consider the taylor’s expansion of $\epsilon_{\theta}(x_t)$, higher-order of moments of the noise $\epsilon$ instead of only the second-order moment will appear in the loss function. Then our results can be extended to that the first-order term in taylor expansion dominates.
>
>
> **Q3: the Hessian of the Lt−1 term is better conditioned when using an informative prior w.r.t. the case of a unit Gaussian prior is also based on the assumption of linearity of the denoising function. But in figure 4, rates of the two curves appear to be similar, just shifted. Do you have any comments on this effect?**
>
> A3: Note that the metric in Fig. 4 is the **logarithm** of the Mel spectrogram mean absolute error (L1 norm). Note that it is the mean absolute error (instead of its logarithm) that is proper to measure the convergence, since it should be measured by a distance of distributions, e.g. the Wasserstein distance, and the Wasserstein distance of two Gaussians is of the same order of the L1 norm of their mean difference. So a shift in the figure indicates a scale of convergence acceleration.
>
>
> **Q4: It would be interesting to compare additional methods from the state of the art.**
>
> A4: Thank you for the feedback about the comparison to the state-of-the-art speech synthesis models. While our initial focus was on discussing and analyzing the efficiency of the diffusion-based model, we acknowledge that the reader will wonder how the final PriorGrad model would stack up against alternative SOTA speech synthesis models. This can show how relevant and impactful the method of PriorGrad will be in a practical sense because our overall focus is indeed in the speech synthesis domain.
>
> Therefore, we have compiled expanded experimental results of PriorGrad with comparison to recent SOTA models from various categories. We have updated the manuscript with Appendix A.6. Several highlights are: **PriorGrad vocoder is the best performing likelihood-based model, and we set a new state-of-the-art acoustic model with PriorGrad**. Please refer to our general response and the updated manuscript for details. We believe that the new results will significantly strengthen our arguments in a practical domain.

---

### Official Review · Reviewer_spyq · 2021-11-01

**Correctness:** 3
**Technical Novelty And Significance:** 3
**Empirical Novelty And Significance:** 2
**Recommendation:** 6
**Confidence:** 3

**Main Review:**

Strengths:
- Novel idea to change the initial sampling distribution to improve inference speed and training convergence.
- Tested for both vocoder and acoustic model.

Weaknesses:

- Mathematical justification of such prior is lacking.
- Same related work is missing.
- Experimental evaluation lacks comparison to alternative models, both for vocoder and acoustic model.
- Baseline acoustic model is non-standard, which makes it even more relevant to compare to other alternative models.
- Acoustic model comparison uses non-optimal vocoder (PWG).
- No discussion and numbers on the inference speed and real-time-factor (RTF), neither for vocoder or acoustic model.
- Code not published.

Details:

Is prior grad mathematically motivated?
"The (DDPM) framework assumes the prior noise as a standard Gaussian distribution"
No, it is not an assumption. It is a mathematical consequence of the definitions of the framework.

Most of the math theory in the paper just shows that convergence rate and training behavior might be better for the proposed method. But it does not give a justification why the data-dependent prior is mathematical sound within the framework.

Related work section seems good, at least related diffusion (or grad-based) models, related prior modeling.It misses many other state-of-the-art vocoder models (MelGAN, UnivNet).

GAN-based vocoders are currently among the best vocoders, such as:

- MelGAN: Generative Adversarial Networks for Conditional Waveform Synthesis (https://arxiv.org/abs/1910.06711)- Multi-band MelGAN: Faster Waveform Generation for High-Quality Text-to-Speech (https://arxiv.org/abs/2005.05106): MelGAN 3.87 MOS, proposed MB-MelGAN 4.22 MOS, Recording 4.58 MOS.
- HiFi-GAN: Generative Adversarial Networks for Efficient and High Fidelity Speech Synthesis (https://arxiv.org/abs/2010.05646): proposed HiFi-GAN V1 4.36 MOS, Recording 4.45 MOS, MelGAN 3.79 MOS, WaveGlow 3.81 MOS, WaveNet (MoL) 4.02 MOS.
- UnivNet: A Neural Vocoder with Multi-Resolution Spectrogram Discriminators for High-Fidelity Waveform Generation (https://arxiv.org/abs/2106.07889): MelGAN: 3.56 MOS, Parallel WaveGAN 3.07 MOS, HiFi-GAN 3.89 MOS, proposed UnivNet-c32 3.93 MOS, Recordings 4.16 MOS.

Those are not compared here.

Then there is WaveGlow, WaveFlow and similar vocoder models which are also not compared.

So experimental validation on vocoder experiments is weak.

Experimental validation on acoustic model is weak as well. Custom-implemented baseline with non-optimal vocoder (PWG). Why not test against other acoustic models such as Diff-TTS or Grad-TTS or FastSpeech 2? Why not use a better vocoder?

Vocoder experiments: LJSpeech. PriorGrad based on DiffWave. T=50 steps for training and inference, for both DiffWave and PriorGrad.
It would be interesting to study the influence of T on both models. Maybe DiffWave with T=100 compensates the effect. Or PriorGrad with T=25. Just some variation on T for both models. You can argue that using the conditional data-dependent prior is an unfair comparison when using the same T.

Acoustic model (TTS) experiment with fixed vocoder:
Table 4 should make more clear what "baseline" is. From the text, it seems to be DiffWave, just as before.
This baseline is also some custom implementation.
Why not use existing implementation, such as Grad-TTS?

Speed (RTF) not evaluated? Not even discussed?

Code?


**Summary Of The Paper:**

This paper presents an extension to conditional denoising diffusion models for text-to-speech (TTS). The initial noise in the inference procedure is not sampled from a standard Gaussian but from a data-dependent conditional non-standard Gaussian, which is called PriorGrad. The motivation is to increase inference speed and training convergence. This method is applied to DiffWave used as a vocoder, and also applied to DiffWave used as an acoustic model (phonemes to mel spect).


**Summary Of The Review:**

The basic idea of using a better initial prior distribution to improve inference speed and training convergence is nice, although it lacks some mathematical justification. The experimental section is lacking too much.

---

> ### Author Response · Authors · 2021-11-23
> **Response to reviewer spyq (part 1/2)**
>
> Thank you for the constructive feedback. We provide a point-to-point response as below:
>
> **Q1: The standard Gaussian it is not an assumption. It is a mathematical consequence of the definitions of the framework.**
>
> A1: Thank you for raising the concern about describing the use of the standard Gaussian in the original DDPM. The reason why we chose the word “assumption” was that the original DDPM paper (Ho et al., NeurIPS 2020) started the formulation with the fixed Markov chain and “chose” the standard Gaussian noise because it satisfies the chained transitions. In other words, the choice of noise can be other types as long as the simplified Markov chain can be achieved. Therefore, we described the method as “assuming” standard or non-standard Gaussian.
>
> We recognize that this may raise confusion as suggested. We will alter the words describing the original DDPM framework by replacing “assumption” with “definition”.
>
> **Q2: it does not give a justification why the data-dependent prior is mathematical sound within the framework.**
>
> A2: We acknowledge that the realization of PriorGrad we presented is a task-specific endeavor, which we already discussed as a limitation of this work in the manuscript. PriorGrad explores methods to provide an approximate proxy of the non-standard Gaussian, based on the conditional information. We do not claim that our methods are faithful to the theoretical formulation (which is, indeed, not possible because one cannot access the optimal mean and variance of the true data distribution).
>
> However, we provided an empirical justification that our approximate proxy of non-standard Gaussian noise is reasonably aligned to the target data, as provided in Sections A.3 and A.5. We hope that these presentations may alleviate the concern, combined with the strong empirical results compared to the baselines and the new experiments with SOTA comparison.
>
> **Q3: It misses many other state-of-the-art vocoder models for comparison.**
>
> A3: The initial scope we presented in this work was discussing and analyzing the efficiency of DDPM, where our experimental claims are verified in the speech synthesis application domain with a controlled and fair experimental design in mind. Therefore, we assumed that presenting the results in the previous diffusion-based model would be sufficient as the controlled experiments.
>
> However, the readers may wonder how the final PriorGrad model would stack up against the previous speech synthesis models, beyond the scope of diffusion-based model as suggested. Therefore, we provide expanded experimental results both on vocoder and the acoustic model, by comparing the final models’ performance with the state-of-the-art approaches. Please refer to the remarks in the bottom.
>
> **Q4: It would be interesting to study the influence of T on both models.**
>
> A4: Thank you for the suggestion that the model performance on different inference steps (T_infer) would be interesting. We took significant care in designing the controlled experiments in the initial manuscript and thought that fixing T will be the most appropriate choice because our major claim is a training acceleration.
>
> Based on the suggestion, we provide extended experimental results with different inference steps from the fully converged models, both on the vocoder and the acoustic model. Please refer to the remarks in the bottom.
>
> **Q5:  Custom-implemented baseline with non-optimal vocoder (PWG). Why not test against other acoustic models such as Diff-TTS or Grad-TTS or FastSpeech 2? Why not use a better vocoder?**
>
> A5: The use of PWG is indeed not an optimal choice for the vocoder given the progress of the high-performance GAN vocoders. In general, the choice of vocoder does not impact the characteristics of the acoustic model itself. And it is a sufficient experimental design to assess the relative performance between each acoustic model as long as the vocoder choice is the same. Therefore, we chose to follow the exact setup provided by FastSpeech 2 (Ren et al., ICLR 2021), where we applied the feed-forward phoneme encoder identical to FastSpeech 2 and used PWG vocoder.
>
> However, we understand that the reader may want to see the acoustic model performance using the state-of-the-art vocoder. We additionally trained HiFi-GAN (Kong et al., NeurIPS 2020)-compatible PriorGrad acoustic models along with the baseline. We provide the additional results compared to the previous SOTA acoustic models, such as FastSpeech 2, Glow-TTS (Kim et al., NeurIPS 2020), and Grad-TTS (Popov et al., ICML 2021). Note that Diff-TTS (Jeong et al., 2021) is not publicly available. Our baseline model’s diffusion decoder is a faithful reproduction of Diff-TTS, but we instead used the feed-forward phoneme encoder of FastSpeech 2. We have prepared an expanded result with the HiFi-GAN vocoder in the updated manuscript.

---

> > ### Author Response · Authors · 2021-11-23
> > **Response to reviewer spyq (part 2/2)**
> >
> > **Q6: The acoustic model baseline is also some custom implementation. Why not use existing implementations, such as Grad-TTS?**
> >
> > A6: By the time we completed the acoustic model experiments, there were no publicly available diffusion acoustic models. Thus, we implemented the custom baseline, which by itself was a non-trivial effort. Furthermore, there is an overlapping contribution of Grad-TTS and our work where both investigated the use of a non-standard Gaussian as the prior, but with different perspectives and realizations. We have documented a detailed comparison between Grad-TTS and PriorGrad in the manuscript: The estimated mean prior (i.e., the encoder’s output) of Grad-TTS is jointly trained with a regression-based auxiliary loss from the target mel-spectrogram. PriorGrad leverages the statistics directly through data, and we do not apply any other objectives to the encoder. Therefore, using Grad-TTS as the baseline conflicts with our approach. We instead provide the comparative results in the updated manuscript, as discussed above.
> >
> > **Q7: Speed (RTF) not evaluated? Not even discussed? Code?**
> >
> > A7: Because the RTF is the same between the baseline and PriorGrad for our controlled experiments, we have not documented it in the initial manuscript. However, we understand that RTF is a main factor of concern in the speech synthesis literature. We have documented RTF for each model in the extended experimental results. We will release the code once the paper is published.
> >
> > **Remarks**
> > We thank the reviewer for suggesting the in-depth analysis of PriorGrad by comparing state-of-the-art methods, along with documenting RTFs which is an important benchmark metric. Based on the feedback, we have prepared significantly expanded experimental results using our fully converged models. We strongly believe that the new results encompassed every feedback: using HiFi-GAN vocoder, varying numbers of the inference step, RTFs, and parameters. In brief, **PriorGrad acoustic model sets a new state-of-the-art** where it performs almost identical to Grad-TTS while being 1.75 times faster. Please refer to our general response and the updated manuscript for details. We hope the reviewer further assesses the experimental results. Thank you again for the feedback.

---

### Official Review · Reviewer_D8iM · 2021-11-03

**Correctness:** 1
**Technical Novelty And Significance:** 3
**Empirical Novelty And Significance:** 2
**Recommendation:** 6
**Confidence:** 4

**Main Review:**


Most of the theory explanations focus on known \mu and \Sigma, but I see very little about how these parameters are estimated and pros and cons about  different estimation techniques.

Relating to point above, can you clarify where is the \mu coming from in case? Noting that original paper had prior as standar normal, you have \mu also. I understand that noise processis zero-mean, but \mu appears in your ELBO. Can you clarify this point.

In 4.1: Why non-clipping variance causes instability in training? Is there a theoretical explanation to it or is it only empirical observation? Relating to  the accompanied footnote, did you try to see if different corpora would have different clipping threshold?

In the Table 1, is GT a the clean, i.e. non-vocoded sample? At least it is supposed to be. And in the same vein, why is 500k training steps better for the proposed than 1M training steps? Can you check statistical significance of this, as if result is significant, then there should be a reason for it.

Intro: 2nd para: "procedurally destroys signal into" ??
In Algorithm 2, line after If-statement seems to have wrong index x_{t-1} to x_t.
Notation seems to be a bit mismatched in style. I would like to see consistent style, for example all vectors to be bolded non-caps and all matrices to be bolded and capitalized. Then it is easier for the reader to follow the story.

Review has been updated based on author rebuttal and discussion.

**Summary Of The Paper:**

In this paper authors propose to use informative prior for conditional diffusion model when applied to neural  vocoding task.

**Summary Of The Review:**

Paper provides quite nice theoretical justifications for their decision to use informative prior instead of classic non-informative one. However, I feel that modification itself is too simple to warrant an ICLR paper. I hope authors can provide a nice rebuttal to this point in assessing the level of contribution of this paper.

The other point that lowers my score is that the final selection of noise variance is based on ad-hoc technique and not justified theoretically, whereas other parts of the paper are.

---

> ### Author Response · Authors · 2021-11-23
> **Response to reviewer D8iM (part 1/2)**
>
> Thank you for the constructive feedback. We provide our point-to-point response below:
>
> **Q1: I see very little about how these parameters are estimated and pros and cons about different estimation techniques. where is the $\mu$ coming from in case?**
>
> A1: We constructed the manuscript starting with the theoretical foundation to the task-specific realization. That is, we presented the theoretical benefits of PriorGrad when we assume that the optimal mean and variance statistics of the data distribution are available. However, this is not possible for any conceivable scenario. PriorGrad explores methods to provide an approximate proxy of such statistics, based on the conditional information.
>
> We would like to highlight that we documented how to estimate the $\mu$ and $\Sigma$ parameters for given tasks in Sections 4.1 and 5.1. In brief: for the vocoder, we computed the prior variance from the spectral energy of the mel-spectrogram. Note that since the waveform data distribution is known to have zero mean, we kept the zero mean prior in the vocoder setup. For acoustic models, we computed the phoneme-level prior mean and variance of the target mel-spectrogram frames.
>
> We also discussed the pros and cons of different techniques: in section A.3, we have discussed different ways to compute the variance with different methods. Also, in section A.5, we discussed many alternative ways to estimate the prior with jointly trainable approaches.
>
>
> **Q2: Why non-clipping variance causes instability in training? Is there a theoretical explanation to it or is it only empirical observation?**
>
> A2: the clipping of minimum standard deviation is an empirical choice to stabilize model training. In our modified ELBO in Prop. 1, there is an inverse of the diagonal variance matrix. If there exists a too-small value (say, zero), its inverse could raise training instabilities (for example, a zero value will cause NaN which leads to training failure). Clipping the minimum of the standard deviation is also previously studied in the waveform synthesis literature (ClariNet, Ping et al., ICLR 2019), which ensures the training stability. The clipping threshold is agnostic to any speech corpora because it serves as a failsafe method to stabilize the model training for handling the corner case with too-low variance.
>
> **Q3:  is GT a the clean, i.e. non-vocoded sample? At least it is supposed to be. And in the same vein, why is 500k training steps better for the proposed than 1M training steps?**
>
> A3: GT (ground-truth) is real audio data, where it provides an upper bound of the human judgment of the speech signal with MOS. Because MOS is subjective, the listeners have different perceptual preferences when assessing the audio quality. PriorGrad with 1M training steps scored the best objective speech metrics. In our auditory tests, the 1M PriorGrad model had even lower background white noise compared to 500K. However, due to this effect, the listener may focus more on other types of artifacts in the sample. We believe this result is from the difference in preference of human perception and the objective metrics. After the initial submission, we found that the 1M training steps model had better performance when we applied fewer inference denoising steps. Therefore, we kept the 1M models for the new expanded experimental results (please refer to the remark section at the bottom).
>
> **Q4: Notation seems to be a bit mismatched in style. I would like to see consistent style, for example all vectors to be bolded non-caps and all matrices to be bolded and capitalized.**
>
> A4: Thank you for raising the consistency concerns in the notation style. We will fix the notations accordingly, as suggested. Please note that the $x_{t-1}$ in Algorithm 2 is correct.

---

> > ### Author Response · Authors · 2021-11-23
> > **Response to reviewer D8iM (part 2/2)**
> >
> > **Q5:  I feel that modification itself is too simple. Noise variance is based on ad-hoc technique and not justified theoretically.**
> >
> > A5: Our primary goal with PriorGrad is a simple and easy-to-use method that will result in a direct impact on applications of the diffusion-based conditional generative models. In other words, PriorGrad offers significant efficiency improvements “for free” without the need to bring potentially overkill approaches like the additionally defined prior encoder or the two-stage training. We believe that the simplicity of PriorGrad holds practical merits on its own, where the application-oriented research community such as speech synthesis strongly favors lightweight approaches for real-world deployments. Furthermore, being able to inject the domain knowledge with our method is significantly helpful for the practical application of the generative model.
> >
> > We emphasize that we have explored many alternatives to realize PriorGrad, such as using other conditional information to compute the prior variance (Section A.3) or applying a jointly trainable estimation of the diffusion prior with an additionally defined estimator (Section A.5). We ultimately found that the “simple” method provided the best results, which we believe is not “bad” per se. The simplicity of our method would enable an adoption of the technique to the existing models.
> >
> > We acknowledge that the realization of PriorGrad we presented is a task-specific endeavor, which we already discussed as a limitation of this work in the manuscript in a transparent way. We do not claim that our methods are faithful to the theoretical formulation (which is, indeed, not possible because one cannot access the mean and variance of the true data distribution). However, we provided an empirical justification that our approximate proxy of non-standard Gaussian noise is reasonably aligned to the target data, as provided in Sections A.3 and A.5. We hope that these presentations may alleviate the concern, combined with the strong empirical results compared to the baselines.
> >
> >
> > **Remarks**
> > We have prepared significantly expanded experimental results with varying degrees of inference steps and comparison with previous SOTA. **PriorGrad also significantly accelerates the inference speed and sets the new state-of-the-art acoustic model.** Please refer to our general response and the updated manuscript (Section A.6) for details. We sincerely hope the reviewer will further assess our additional experimental results and the level of contribution of the work. Thank you again for the feedback.

---

> > ### Comment · Reviewer_D8iM · 2021-11-23
> > **Thanks for the rebuttal, and one quick comment.**
> >
> > Thanks a lot for your rebuttal, it clarified a lot.
> >
> > I am thinking that authors answer to Q3 is not consistent. It is absolutely true MOS is a perceptual metric and of course subject to listeners biases and multiple random effects. What I was concerned about is that is there something in the estimation / training procedure that resulted in objective difference in output between 500k and 1M steps. I feel that last part of your answer seems to indicate that there is an objective difference. So my original question still stands.

---

> > > ### Author Response · Authors · 2021-11-29
> > > **Additional clarification on Q3**
> > >
> > > Thank you very much for raising the additional question regarding the subjective score of the 500K vs. 1M checkpoints of our method. We would like to further clarify our observations and hope the reviewer to alleviate the concern:
> > >
> > > Our previous response was based on the author's self-listening tests. During the additional evaluations with varying inference steps, we observed that the overall quality of the sample was noticeably better for models trained for longer durations (1M steps). This was true for both the baseline DiffWave model and our PriorGrad model. In general, diffusion-based models increase in performance for various sampling steps if we increase the training compute for longer durations (IDDPM, Nichol et al., ICML 2021). The difference is more identifiable for smaller inference steps. In other words, the difference between the models becomes smaller if we employ larger inference steps.
> > >
> > > To support our observation to scrutiny, we have conducted an additional set of MOS evaluations for the vocoder models. We used the previously used 500K and 1M checkpoints and applied T_infer = 6 or T_infer=50 for both models:
> > >
> > > | Method | Training steps | T_infer | MOS |
> > > | :---: | :---: | :---: | :---: |
> > > | GT | - | 4.42 ± 0.07 | - |
> > > | DiffWave | 500K | 6 | 3.98 +- 0.08 |
> > > | DiffWave | 500K | 50 | 4.12 +- 0.08 |
> > > | DiffWave | 1M | 6 | 4.01 +- 0.08 |
> > > | DiffWave | 1M | 50 | 4.12 +- 0.08 |
> > > | PriorGrad | 500K | 6 | 4.02 +- 0.08 |
> > > | PriorGrad | 500K | 50 | 4.21 +- 0.07 |
> > > | PriorGrad | 1M | 6 | 4.14 +- 0.08 |
> > > | PriorGrad | 1M | 50 | 4.25 +- 0.07 |
> > >
> > > From the result above, we observed that:
> > >
> > > 1. For both models, T_infer=50 was significantly better than T=_infer=6.
> > >
> > > 2. For T_infer=6, PriorGrad trained for 1M steps was also significantly better than the 500K checkpoint.
> > >
> > > 3. For T_infer=50, Baseline Diffwave showed no measurable difference between 500K and 1M steps, consistent with the previous results.
> > >
> > > 4. For T_infer=50, PriorGrad's performance on 1M steps was improved, but the relative difference between 500K and 1M is less emphasized. This time, PriorGrad with 1M steps scored better than 500K steps. However, the relative difference is smaller than other configurations and lies within the confidence interval. This suggests that the T_infer=50 performance difference of PriorGrad between 500K and 1M steps is not as significant as other configurations.
> > >
> > > From the author's listening tests, PriorGrad (with T_infer=50) with 1M steps had slightly lower white noise compared to 500K steps. Also, the high-frequency peak noise for several segments (such as during sibilance) seems to be slightly more emphasized for the model with 1M steps. Please note that we have not claimed that the PriorGrad vocoder trained for 500K steps had globally better performance compared to 1M steps from the manuscript. What we claimed in this work is that PriorGrad performed significantly better than the baseline DiffWave under fewer training steps, or under fewer inference steps from updated experiments.
> > >
> > > We hope that the additional clarification, along with the additional set of MOS tests, would alleviate the reviewer's remaining questions. Thank you again for the additional feedback.

---

> > > > ### Comment · Reviewer_D8iM · 2021-11-29
> > > > **Authors have clarified my concerns**
> > > >
> > > > I feel that this additional experiment to Q3 is now sufficient and it shows the difference between DiffWave and the proposed.
> > > >
> > > > The key missing point in the paper is still the more theoretical explanation of Q1, i.e. where the \mu is coming from.
> > > >
> > > > However, I am willing to raise my score by one point.

---

### Author Response · Authors · 2021-11-23
**General response to all reviewers**

We would like to thank all reviewers for the constructive feedback to improve the quality of the paper. Especially, reviewer spyq and Wdw2 suggested **further assessment of the proposed method**, such as using a varying number of inference denoising steps and comparing the final model to state-of-the-art speech synthesis models. We deliberately prepared expanded experimental results of PriorGrad based on the feedback. We sincerely hope that these additional results strengthen the level of contribution of the work, as also suggested by other reviewers (D8iM, 518v).

We have updated the manuscript with the new experimental results of PriorGrad in appendix A.6., where we further analyze the performance with different numbers of inference steps and comparison to state-of-the-art. Here, we highlight several new achievements of PriorGrad:

* PriorGrad vocoder with 6 inference steps matches the performance of the baseline DiffWave with the full 50 steps. This suggests that the informative prior also significantly **accelerates the inference**, which is a major drawback of the diffusion-based generative models.

* We have switched the acoustic model’s pretrained vocoder to HiFi-GAN (Kong et al., NeurIPS 2020) for comparison to previous work. **PriorGrad acoustic model sets a new state-of-the-art**, where our model outperforms all alternative methods. Furthermore, it performs almost identical to a concurrent work, Grad-TTS (Popov et al., ICML 2021), while being approximately 1.75 times faster.

* PriorGrad acoustic model is also robust to an extreme case with **just two inference denoising steps**, where the speed gets close to the feed-forward model while keeping higher quality.

Please refer to the below tables for details:


### Expanded vocoder results

| Method | T_infer | MOS | RTF | Params |
| :---: | :---: | :---: | :---: | :---: |
| GT | - | 4.60 ± 0.05 | - | - |
| DiffWave/PriorGrad | 6 | 4.10 ± 0.08 / 4.20 ± 0.08 | 0.1388 |  2.62M |
|  | 12 | 4.15 ± 0.08 / 4.29 ± 0.08 | 0.2780 |  |
|  | 50 | 4.19 ± 0.07 / **4.33 ± 0.07** | 1.1520 |  |
| WaveGlow | - | 4.09 ± 0.08 | 0.0780 |  87.9M |
| WaveFlow | - | 4.01 ± 0.09 | 0.1759 |  22.3M |
| HiFi-GAN (V1) | - | **4.44 ± 0.05** | **0.0068** |  14.0M |


### Expanded acoustic model results

|       Method       | T_infer |            MOS            |   RTF  | Params(Enc) | Params(Dec) |
|:---:|:---:|:---:|:---:|:---:|:---:|
|         GT         |    -    |        4.65 ± 0.05        |    -   |      -      |      -      |
|    GT (Vocoder)    |    -    |        4.50 ± 0.06        |    -   |      -      |      -      |
| Baseline/PriorGrad |    2    | 2.80±0.17/4.25±0.08 | **0.0069** |    11.5M    |     3.5M    |
|                    |    6    | 3.67±0.12/4.29±0.07 | 0.0113 |     |     |
|                    |    12   | 4.14±0.08/**4.39±0.08** | 0.0176 |   |    |
|      Grad-TTS      |    2    |        3.43±0.15        | 0.0090 |     7.2M    |     7.6M    |
|                    |    10   |        **4.38 ± 0.05**        | 0.0308 |    |    |
|     FastSpeech2    |    -    |        4.19 ± 0.08        | **0.0040** |    11.5M    |    11.5M    |
|      Glow-TTS      |    -    |        4.23 ± 0.08        | 0.0081 |     7.2M    |    21.4M    |

---

### Decision · Program_Chairs · 2022-01-20

**Decision:**

Accept (Poster)

**Comment:**

This paper suggests using a conditional prior in conditional diffusion-based generative models. Typically, only the score function estimator is provided with the conditioning signal, and the prior is an unconditional standard Gaussian distribution. It is shown that making the prior conditional improves results on speech generation tasks.

Several reviewers initially recommended rejection, but after extensive discussion and interaction with the authors, all reviewers have given this work a "borderline accept" rating.

Criticisms included that the idea is too simple or obvious to warrant an ICLR paper. I am inclined to disagree: simple ideas that work are often the ones that persist and see rapid adoption (dropout regularisation is my favourite example). Like the authors, I believe simplicity is an advantage in this respect, rather than a disadvantage. Of course, simple ideas do require extensive and convincing empirical validation to be worth publishing at ICLR. After the authors' updates, I believe the work meets this bar.

Another issue raised by several reviewers is the limited theoretical justification for the approach. However, combined with the simplicity of the method, I believe the empirical results of the revised version sufficiently justify the approach on their own. Nevertheless, I would recommend that the authors consider further how they could address this issue in the final version of their manuscript, as they have already begun to do during the discussion phase.

Another way to strengthen the paper further would be to demonstrate how the generic approach can be applied in a different domain (e.g. conditional image generation), but I do not consider this addition necessary for the work to warrant publication.

In light of this, I am recommending acceptance.